# Glutamine deficiency in solid tumor cells confers resistance to ribosomal RNA synthesis inhibitors

Melvin Pan[1], Christiane Zorbas[2], Maki Sugaya[1], Kensuke Ishiguro[3,4], Miki Kato[1], Miyuki Nishida[1], Hai-Feng Zhang[5,6], Marco M. Candeias [7], Akimitsu Okamoto [3], Takamasa Ishikawa[8], Tomoyoshi Soga [8], Hiroyuki Aburatani [9], Juro Sakai[10,11], Yoshihiro Matsumura [10,11], Tsutomu Suzuki [3], Christopher G. Proud[12,13], Denis L. J. Lafontaine [2] & Tsuyoshi Osawa [1,3✉]

Ribosome biogenesis is an energetically expensive program that is dictated by nutrient availability. Here we report that nutrient deprivation severely impairs precursor ribosomal RNA (pre-rRNA) processing and leads to the accumulation of unprocessed rRNAs. Upon nutrient restoration, pre-rRNAs stored under starvation are processed into mature rRNAs that are utilized for ribosome biogenesis. Failure to accumulate pre-rRNAs under nutrient stress leads to perturbed ribosome assembly upon nutrient restoration and subsequent apoptosis via uL5/uL18-mediated activation of p53. Restoration of glutamine alone activates p53 by triggering uL5/uL18 translation. Induction of uL5/uL18 protein synthesis by glutamine is dependent on the translation factor eukaryotic elongation factor 2 (eEF2), which is in turn dependent on Raf/MEK/ERK signaling. Depriving cells of glutamine prevents the activation of p53 by rRNA synthesis inhibitors. Our data reveals a mechanism that tumor cells can exploit to suppress p53-mediated apoptosis during fluctuations in environmental nutrient availability.

[1] Division of Integrative Nutriomics and Oncology, RCAST, The University of Tokyo, 4-6-1 Komaba, Meguro-ku, Tokyo 153-8904, Japan. [2] RNA Molecular Biology, Fonds de la Recherche Scientifique (F.R.S.-FNRS), Université Libre de Bruxelles (ULB), Biopark campus, 6041 Gosselies, Belgium. [3] Department of Chemistry and Biotechnology, Graduate School of Engineering, The University of Tokyo, Tokyo 113-8656, Japan. [4] RIKEN Center for Biosystems Dynamics Research, 1-7-22 Suehiro-cho, Tsurumi-ku, Yokohama, Kanagawa 230-0045, Japan. [5] Department of Molecular Oncology, BC Cancer, Vancouver, BC V5Z 1L3, Canada. [6] Department of Pathology and Laboratory Medicine, University of British Columbia, Vancouver, BC V6T 2B5, Canada. [7] Molecular and RNA Cancer Unit, Graduate School of Medicine, Kyoto University, Kyoto, Japan. [8] Institute for Advanced Biosciences, Keio University, Tsuruoka 997-0052, Japan. [9] Genome Science Division, RCAST, The University of Tokyo, 4-6-1 Komaba, Meguro-ku, Tokyo 153-8904, Japan. [10] Division of Metabolic Medicine, RCAST, The University of Tokyo, 4-6-1 Komaba, Meguro-ku, Tokyo 153-8904, Japan. [11] Division of Molecular Physiology and Metabolism, Tohoku University Graduate School of Medicine, Sendai 980-8574, Japan. [12] Lifelong Health, South Australian Health & Medical Research Institute, Adelaide, SA 5000, Australia. [13] School of Biomedical Sciences, University of Adelaide, Adelaide, SA 5005, Australia. ✉email: osawa@lsbm.org

Solid tumors outstrip their blood supply and develop tissue microenvironments that are depleted of oxygen and nutrients[1,2]. Such microenvironments are characterized by chronic hypoxia and are typically located more than 180 μm away from blood vessels[3,4]. Another type of hypoxia—known as 'cycling hypoxia'—arises from transient shutdown of immature vasculature and can lead to 'reoxygenation injury', where increased free radical synthesis causes oxidative stress and tissue damage[5,6]. Along with hypoxia, an unstable blood supply also causes substantial fluctuations in nutrient availability in the tumor microenvironment[7]. Nutrient deprivation severely inhibits tumor cell proliferation but selects for aggressive cells that display increased angiogenic and metastatic ability[8,9].

In order to preserve energy balance during metabolic stress, tumor cells evolve adaptive mechanisms to attenuate ATP-costly processes under nutrient deprivation[10–12]. These adaptations involve the inhibition of protein synthesis, which is the most energy-consuming process in the cell[12]. For instance, the eukaryotic elongation factor 2 kinase (eEF2K) is activated under nutrient restriction and inhibits the translation elongation factor eEF2, thereby suppressing overall protein synthesis and promoting cell survival under nutrient depletion[2]. Metabolic stress inhibits mammalian target of rapamycin complex 1 (mTORC1)[13]. This impairs mRNA translation initiation by activation of the eIF4E-binding protein 1 (4E-BP1), which inhibits the cap-binding translation initiation factor eIF4E[13]. Nutrient stress also triggers the integrated stress response (ISR), which leads to the inactivation of the translation factor eIF2α, thereby inhibiting translation initiation and promoting the synthesis of stress-adaptive proteins including ATF4[14]. Another critical adaptation is the suppression of rRNA synthesis by RNA Polymerase I (Pol I), which comprises up to 60% of total cellular transcription[15,16]. mTORC1 and AMP-activated protein kinase (AMPK) reciprocally regulate pre-rRNA biosynthesis by phosphorylating TIF-IA, the protein complex that anchors Pol I to the rDNA promoter[15,16]. While oncogenic adaptations to nutrient deprivation have been well characterized, it is unknown how starving tumor cells resume proliferative capacities upon nutrient restoration.

Ribosome biogenesis is a highly sophisticated pathway that requires the production of four rRNAs and eighty ribosomal proteins[17,18]. Hundreds of protein trans-acting factors and small nucleolar RNAs aid the assembly process[17,18]. The process starts in the nucleolus where Pol I synthesizes a long polycistronic pre-rRNA that undergoes rapid folding, modification, processing, and associations with ribosomal proteins[19,20]. In unstressed cells, the tumor suppressor p53 is maintained at low levels by constitutive proteasomal degradation promoted by MDM2 (HDM2 in humans, henceforth denoted MDM2)[21]. When ribosome biogenesis is compromised, ribosomal components freely accumulate, including the ribosomal proteins uL5 (formerly known as RPL11) and uL18 (RPL5), which, together with the 5S rRNA, form a stable trimeric complex that binds and inhibits MDM2, leading to p53 stabilization and subsequent p53-mediated cell cycle arrest or apoptosis[22–24].

Given the importance of ribosome biogenesis in cancer cell growth, highly potent first-in-class Pol I inhibitors—such as CX-5461 and BMH-21—have been developed to block rRNA synthesis and stabilize p53[25,26]. These inhibitors have the advantage of stabilizing p53 without inducing severe DNA damage, which is beneficial over mainstay genotoxic chemotherapies. Surprisingly, despite the importance of ribosome biogenesis to all cell types, single-agent therapy with CX-5461 has proven more effective against p53 wild-type hematological malignancies than against p53 wild-type solid tumors[25,27–29]. Notably, CX-5461 has been shown to possess antitumor activity against select solid tumor cell lines[30]. However, given the lack of data demonstrating in vivo on-target drug activity (pre-rRNA depletion, p53 stabilization), it is not entirely clear whether the antitumor activity is due to on-target or secondary effects.

Here, we report that nutrient deprivation severely impairs pre-rRNA processing and causes the accumulation of unprocessed rRNAs. Upon nutrient refeeding, protein synthesis resumes and newly synthesized ribosomal proteins (including uL5/uL18) associate with stored pre-rRNAs for ribosome assembly. Failure to accumulate pre-rRNAs under nutrient starvation leads to cell-lethal p53 stabilization, which is caused by uL5/uL18-mediated inhibition of MDM2. We also show that the activation of p53 by uL5/uL18 is dependent on glutamine availability. Glutamine mediates the production of unassembled uL5/uL18 by activating Ras/Raf/MEK/ERK signaling, which suppresses eEF2K to allow global eEF2-mediated translation. Our study identifies pre-rRNA accumulation as a mechanism by which cells use to stay alive during fluctuations in environmental nutrient availability. Finally, we show that glutamine deprivation prevents the activation of p53 by rRNA synthesis inhibitors, strongly suggesting that glutamine-deficient solid tumors display inherent resistance to small molecules that target Pol I.

## Results

**Nutrient deprivation upregulates pre-rRNA expression.** We first sought to understand how nutrient stress affects the rate of pre-rRNA synthesis by Pol I. We used the 5-fluorouridine (5FU) labeling assay to measure nascent pre-rRNA synthesis (see "Methods"). 5FU is a nucleoside analog that is incorporated into nascent RNAs, however, as the majority of newly synthesized RNA is pre-rRNA, 5FU labeling mainly occurs in the nucleolus (Supplementary Fig. 1a). Of note, this assay measures the rate of pre-rRNA synthesis (i.e. Pol I activity) rather than total pre-rRNA expression. To study nutrient stress, a panel of cell lines (colorectal HCT116, melanoma A375, lung adenocarcinoma A549, osteosarcoma U2OS, gastric MKN45) were cultured in medium lacking glucose, amino acids, and serum. After 24 h of starvation, the cells were pulsed with 5FU and processed for immunofluorescence. 5FU incorporation was significantly suppressed by nutrient deprivation (ND) in all cell lines tested (Fig. 1a). Thus, Pol I-mediated pre-rRNA synthesis is abrogated by metabolic stress.

We next tested the effect of ND on the expression of each pre-rRNA species (Fig. 1b). HCT116 cells were subjected to ND for 24 h and pre-rRNA expression was analyzed by northern blotting. The Pol I inhibitor CX-5461 was used as a positive control for the inhibition of pre-rRNA synthesis. As expected, CX-5461 swiftly depleted the expression of all pre-rRNA intermediates within 2 h (Fig. 1c). This acute effect is due to the short half-lives of the pre-rRNAs (approximately 30 min), which is in turn caused by rapid processing[30]. Surprisingly, despite the inhibitory effect of ND on pre-rRNA synthesis, the expression of large precursors (47S, 45S) did not decrease after 24 h ND (Fig. 1c). ND also induced the robust upregulation of the downstream 30S species by approximately two-fold (Fig. 1c). Other precursors (21S, 18S-E) were slightly downregulated after 24 h of ND but were not decreased to the extent observed in CX-5461 treated cells (Fig. 1c, Supplementary Fig. 1b). Similarly, in mouse 3T3-L1 fibroblasts, ND did not affect 47S/45S levels and upregulated the 34S intermediate (equivalent to human 30S) (Fig. 1d). Taken together, these results show that suppressed pre-rRNA synthesis under ND does not lead to decreased pre-rRNA expression.

We next used qRT-PCR to quantify the effect of ND on pre-rRNA levels. Given that the 30 S intermediate is upregulated by ND (Fig. 1c), we selected primers that measure the total

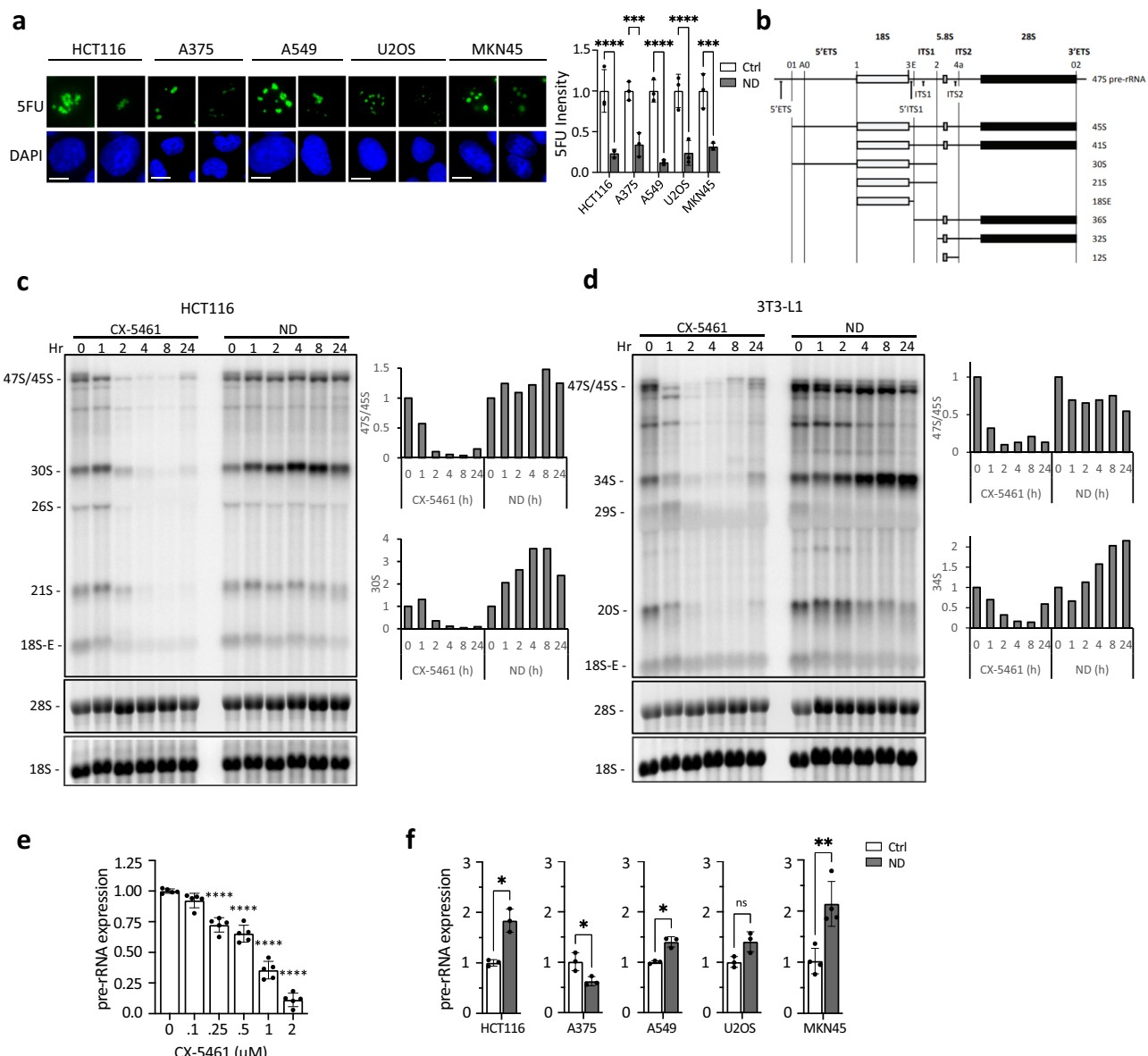

**Fig. 1 Nutrient deprivation upregulates pre-rRNA expression. a** 5FU labeling under nutrient deprivation (24 h). Cancer cell lines were cultured under ND (-Glucose–Amino acids–FBS) for 24 h, and pre-rRNA synthesis was measured by 5FU metabolic labeling. Data represent the mean ± SD of three images. *P* values (left to right): <0.0001, $2 \times 10^{-4}$, <0.0001, <0.0001, 0.0001 (∗∗∗*p* < 0.001; ∗∗∗∗*p* < 0.0001, statistical analysis by two-way ANOVA). Scale bar, 10 μm. **b** Schematic of pre-rRNA processing. 47S processing produces intermediate precursors which mature to 18S, 28S, and 5.8S rRNAs. The primary 47S pre-rRNA contains two external transcribed spacers (5′ETS and 3′ETS) and two internal transcriber spacers (ITS1 and ITS2). **c** Left: Pre-rRNA expression in HCT116 treated to CX-5461 or ND using northern blotting (ITS1 Probe). Right: Quantification of 47S/45S and 30S intermediates in response to CX-5461(10 μM) /ND. The full northern blot is shown in Supplementary Fig. 1b. **d** Left: Pre-rRNA expression in 3T3-L1 cells treated to CX-5461 or ND using northern blotting (ITS1 Probe). Right: Quantification of 47S/45S and 34S intermediates in response to CX-5461/ND. The full northern blot is shown in Supplementary Fig. 1c. **e** Expression of pre-rRNAs (47S/45S/30 S species) after CX-5461 (10 μM, 8 h) exposure. Data represent mean ± SD of *n* = 5 technical replicates. *P* values (left to right): <0.0001, <0.0001, <0.0001, <0.0001 (∗∗∗*p* < 0.0001, statistical analysis by one-way ANOVA). **f** Expression of pre-rRNAs (47S/45S/30S species) after 24 h ND in HCT116, A549, U2OS, and MKN45 cells, as assessed using qRT-PCR. Data represent mean ± SD of *n* = 3 technical replicates. *P* values (left to right): 0.0132, 0.0213, 0.0115, 0.0040 (∗*p* < 0.05; ∗∗*p* < 0.01, statistical analysis by two-way ANOVA).

abundance of the 47S, 45S, and 30S precursors. These primers bind across a cleavage site of the 5′-external transcribed sequence (5′ETS) of the pre-rRNA (Supplementary Fig. 1d). CX-5461 significantly decreased pre-rRNA expression (Fig. 1e). In contrast, 24 h ND significantly upregulated pre-rRNA levels in HCT116, A549, and MKN45 cells (Fig. 1f).

**Pre-rRNA processing is impaired under ND.** We hypothesized that upregulated pre-rRNA levels under ND is a direct

consequence of impaired pre-rRNA processing. We used the L-(methyl-$^3$H)-methionine metabolic labeling assay to assess pre-rRNA processing[31]. Control and ND HCT116 cells were pulsed with L-(methyl-$^3$H)-methionine for 30 min and then chased with excess cold methionine (Fig. 2a). The maturation of L-(methyl-$^3$H)-methionine labeled pre-rRNAs was tracked over 4 h. In control cells, the expression of large precursors (47S, 45S) and downstream intermediates (32S, 30S) quickly decreased over time, indicating that these precursors were metabolized into

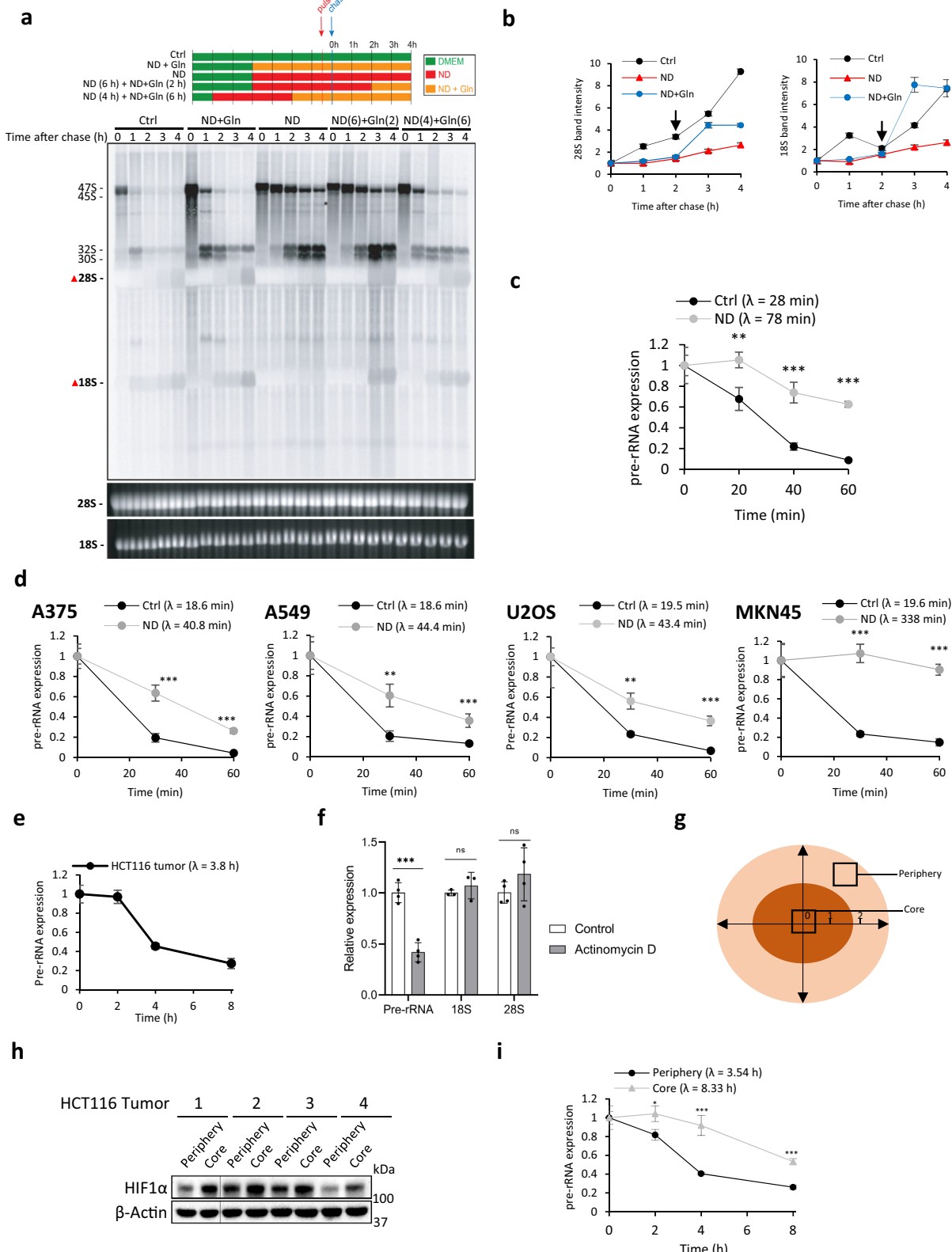

mature 18S/28S rRNAs (Fig. 2a). In contrast, ND caused the robust accumulation of the 47S, 45S, 32S, and 30S precursors (Fig. 2a), demonstrating that these intermediates did not mature to 18S/28S rRNAs. Accordingly, the production of 18S and 28S rRNAs was strongly impaired under ND (Fig. 2b). Remarkably, adding back the non-essential amino acid glutamine alone to starved cells fully rescued 18S and 28S rRNA production

(Fig. 2b). This indicates that unprocessed rRNAs that accumulate under ND are utilized for ribosome biogenesis during nutrient restoration.

To rigorously confirm that ND slows pre-rRNA processing, we measured the pre-rRNA half-life using qRT-PCR and the 5′ETS primers (Supplementary Fig. 1d). Transcription was blocked in control and ND HCT116 cells using actinomycin D (20 µM), and

**Fig. 2 Pre-rRNA processing is impaired under ND. a** Analysis of pre-rRNA processing kinetics in control or ND HCT116 cells. HCT116 cells that were pretreated to 3.5 h ND were subsequently pulse labeled with L-(methyl-3H)-methionine for 30 min and chased with excess cold methionine. To assess the effect of nutrient restoration on pre-rRNAs accumulated under ND, Gln was added back to the ND medium 2 h after the cold methionine chase ("ND + Gln"). **b** Quantification of the 28S/18S bands from Fig. 1f (as indicated by the red triangles). The black arrow at the 2 h time point indicates the time at which glutamine was added to starved (ND) cells. **c** ND increases the pre-rRNA half-life ($\lambda$) as assessed using qRT-PCR. HCT116 cells were pretreated to (24 h) ND, transcription was stopped by (20 µM) actinomycin D, and total RNA at the indicated time points for qRT-PCR analysis. Data represent mean ± SD of $n = 3$ technical replicates. P values (left to right): 0.006, 0.0007, $3.2 \times 10^{-6}$ ($*p < 0.05$; $**p < 0.01$; $***p < 0.001$). **d** ND slows pre-rRNA processing in A375, A549, U2OS, MKN45 cells. Data represent mean ± SD of $n = 3$ technical replicates. P values (left to right): A375, 0.00081, $7.2 \times 10^{-6}$, A549, 0.0035, $9.4 \times 10^{-5}$, U2OS, 0.0015, 0.0031, MKN45, 0.00025, 0.00031 ($*p < 0.05$; $**p < 0.01$; $***p < 0.001$). **e** HCT116 tumor pre-rRNA half-life as determined using qRT-PCR. Data represent mean ± SD of $n = 3$ technical replicates. **f** Expression of pre-rRNAs, 18S, and 28S in HCT116 tumors after actinomycin D (1.33 mM) intratumoral injection. Data represent mean ± SD of $n = 4$ technical replicates. P values (left to right): $6.5 \times 10^{-5}$, 0.20, 0.12 (ns, not significant; $*p < 0.05$; $**p < 0.01$; $***p < 0.001$). **g** Schematic of core and periphery regions of established HCT116 tumors (approximate size 1000 mm³). **h** Expression of HIF-1α in HCT116 periphery and core tissues as assessed using western blotting. **i** Pre-rRNA processing is faster in peripheral tissues compared to core tissues, as determined using qRT-PCR. Data represent mean ± SD of $n = 3$ technical replicates. P values (left to right): 0.046, 0.00064, 0.00016 ($*p < 0.05$; $**p < 0.01$; $***p < 0.001$).

pre-rRNA expression was monitored over 1 h (see "Methods"). In control HCT116 cells, the pre-rRNA half-life was approximately 28 min, while cells subjected to ND displayed a significantly longer half-life of 78 min (Fig. 2c). Similarly, ND significantly increased the pre-rRNA half-life by at least two-fold in A375, A549, U2OS, and MKN45 cells (Fig. 2d).

Solid tumors are insufficiently vascularized with microenvironments that are strongly deprived of nutrients[7]. We therefore sought to estimate the half-life of pre-rRNAs in vivo. To investigate this, we injected a highly concentrated solution of actinomycin D (1.33 mM) directly into HCT116 tumors to acutely halt transcription. After injection, tumor tissues were found to contain >100 µM Actinomycin D, indicating sufficient drug diffusion (Supplementary Fig. 1e). Further, since the inhibition of Pol I transcription causes nucleolar proteins to translocate to the nucleoplasm, a process known as nucleolar disruption[32,33], we assessed the localization of nucleolin in tumor cryosections. Indeed, control tumors exhibited many intact nucleolin-positive nucleoli, but actinomycin D-treated tumors displayed nucleolin staining in the cytoplasm (Supplementary Fig. 2a). Next, to assess the stability of pre-rRNAs in vivo, HCT116 tumors were intratumorally injected with actinomycin D and tissues were collected over 8 h. RNA was extracted from the tissues and pre-rRNA expression was analyzed by qRT-PCR. Tumoral pre-rRNAs displayed a half-life of approximately 3.8 h (Fig. 2e). Further, actinomycin D had no significant effect on 18S and 28S expression (Fig. 2f). Next, since it has been shown that the cores of solid tumors are more severely depleted of nutrients compared to the periphery[34], we reasoned that core tumor cells would display slower pre-rRNA processing kinetics than cells in the periphery. To test this, large HCT116 tumors (>1000 mm³) were injected with actinomycin D and peripheral and core tissues were dissected to differentiate tumor cells under high or low nutrient conditions, respectively (Fig. 2g)[34]. Core tissues expressed higher levels of HIF-1α, consistent with the notion that core regions of solid tumors are hypoxic (Fig. 2h). Furthermore, metabolomic analysis showed that core tissues had significantly lower levels of several metabolites including glutamine, arginine, serine, and histidine (Supplementary Fig. 2e). Finally, in line with our hypothesis, peripheral pre-rRNAs exhibited a half-life of 3.5 h while core pre-rRNAs displayed a significantly longer half-life of 8.3 h (Fig. 2i).

**Pre-rRNA accumulation is an adaptive response to nutrient restoration.** Since pre-rRNAs that accumulate under ND are utilized for ribosome biogenesis during nutrient (glutamine) feedback (Fig. 2b), we hypothesized that pre-rRNA accumulation may be important to the resumption of ribosome assembly during

nutrient recovery. To address this, we tested whether inhibiting pre-rRNA expression under ND leads to perturbed ribosome assembly during metabolic recovery. First, to prevent pre-rRNA accumulation, HCT116 or A375 cells were placed under ND with co-treatment of vehicle, CX-5461, or BMH-21 (a second Pol I inhibitor). Inhibition of pre-rRNA synthesis with CX-5461 or BMH-21 effectively blocked pre-rRNA accumulation under ND (Fig. 3a). Notably, ND alone caused high cell death, but this was not exacerbated by CX-5461 or BMH-21 co-treatment (Fig. 3b). Next, to induce metabolic recovery, the 'pre-rRNA-depleted' cells (ND + CX-5461 or ND + BMH-21) were transferred to drug-free nutrient-rich media for 8 h (Fig. 3c). The fidelity of ribosome assembly was evaluated by assessing the expression of unassembled ribosomal proteins, which accumulate upon defective ribosome biogenesis[35]. To evaluate ribosome-free proteins, protein lysates were depleted of large ribosomes using sucrose cushion ultracentrifugation, and the resulting ribosome-free lysates were analyzed by western blotting (Fig. 3d). We focused on the expression of the key ribosomal proteins uL5 (formerly RPL11) and uL18 (RPL5) because these two proteins have been shown to specifically accumulate in the ribosome-free fraction upon impaired ribosome biogenesis[35]. Ribosome-free uL5/uL18 was undetectable in ND vehicle cells subjected to nutrient restoration (NR), indicating that newly translated uL5/uL18 were efficiently incorporated into ribosomes during metabolic recovery (Fig. 3e). In contrast, unassembled uL5/uL18 was expressed in pre-rRNA-depleted cells (ND + CX-5461) subjected to NR (Fig. 3e). This showed that these ribosomal proteins were not incorporated into ribosomes. Moreover, halting translation with cycloheximide (CHX) prevented the accumulation of unassembled uL5/uL18 (Fig. 3e). This is consistent with the proposed model that ongoing protein synthesis is required for the accumulation of unassembled uL5/uL18[35]. We further assessed whether perturbed ribosome assembly was associated with nucleolar disruption. Indeed, nucleolin was found in the nucleoplasm in pre-rRNA-depleted cells exposed to NR, whereas the staining was confined to the nucleolus in control cells (Supplementary Fig. 3a).

To assess the consequence of impaired ribosome assembly, cells depleted of pre-rRNAs were exposed to prolonged NR (24 h) and cell recovery was initially assessed using light microscopy. Starved control (ND + Vehicle) HCT116 and A375 cells resumed proliferation after NR, as shown by increased confluence. In striking contrast, starved pre-rRNA-depleted cells exposed to NR did not resume proliferation and further exhibited features of apoptosis such as reduced cell size and rounded morphology (Fig. 3f). Trypan blue exclusion assays revealed profound levels of cell death in pre-rRNA-depleted cells undergoing NR but not in control cells (Fig. 3g). The presence of cleaved caspase-3 and

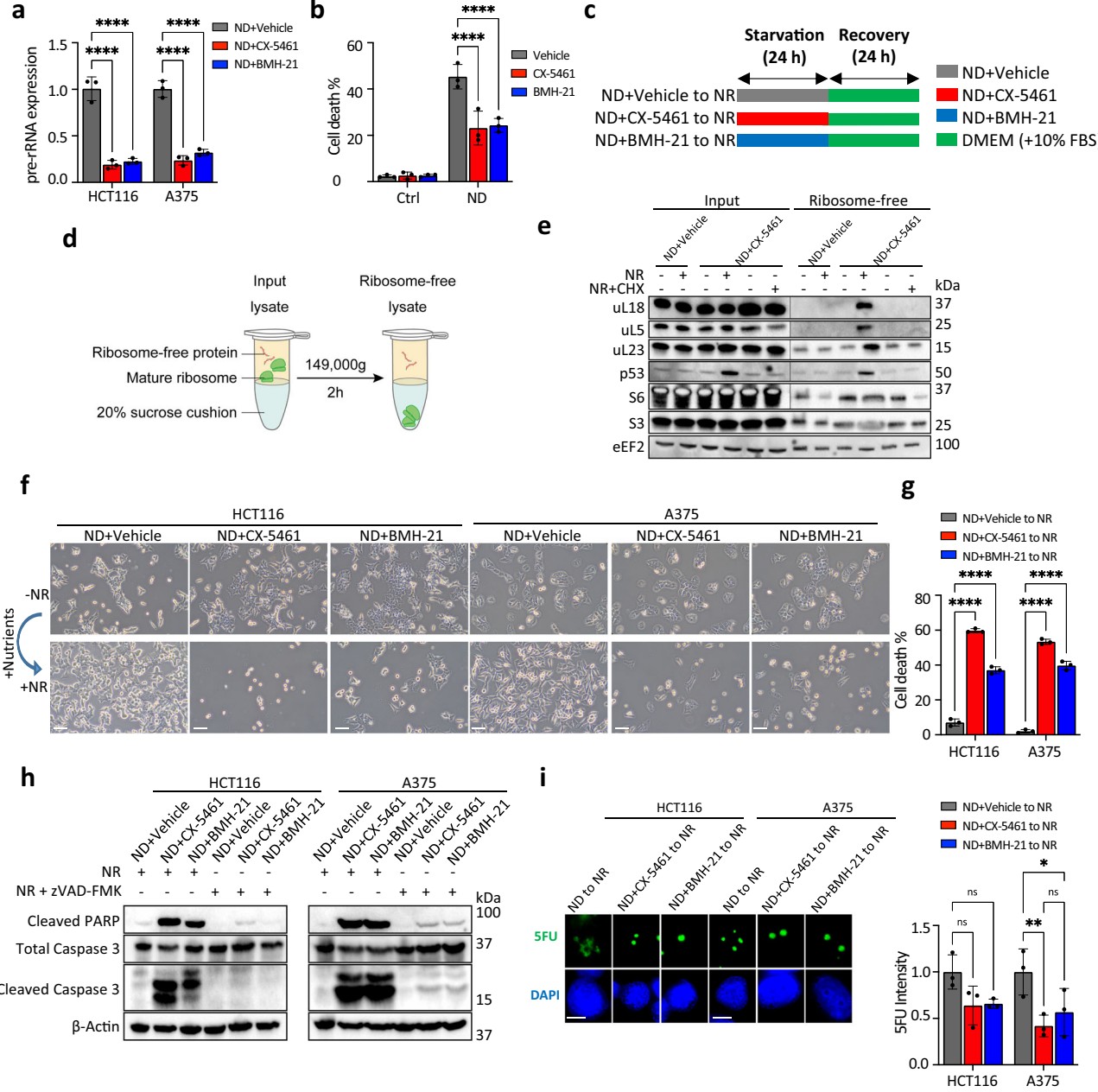

**Fig. 3 Pre-rRNA accumulation is an adaptive response to nutrient restoration. a** Pre-rRNA expression after treatment with vehicle (NaH$_2$PO$_4$), CX-5461 (10 µM), or BMH-21 (1 µM) under 24 h ND. Data represent mean ± SD of $n = 3$ technical replicates. $P$ values (left to right): HCT116, <0.0001, <0.0001, A375, <0.0001, <0.0001, statistical analysis by two-way ANOVA. **b** Percentage of cell death in HCT116 cells cultured in control or ND medium with vehicle (NaH$_2$PO$_4$), CX-5461 (10 µM), or BMH-21 (1 µM) as assessed using trypan blue exclusion assay. Data represent mean ± SD of $n = 3$ biological replicates. (ns: not significant). $P$ values (left to right): ND, <0.0001, <0.0001, statistical analysis by two-way ANOVA. **c** Schematic showing ND and nutrient restoration (NR) treatments. To inhibit pre-rRNA expression, HCT116 or A375 cells were first placed under ND in the presence of vehicle, CX-5461, or BMH-21. Cells pretreated to ND + CX-5461 (10 µM)/ND + BMH-21 (1 µM) represent "pre-rRNA-depleted" cells. To induce metabolic recovery, pre-starved cells were washed three times with PBS and were allowed to rest in drug-free DMEM (10% FBS). **d** In order to assess the expression of ribosome-free proteins, input lysates were clarified of large ribosomes using sucrose cushion ultracentrifugation. The resulting ribosome-free lysates were analyzed using western blotting. **e** Expression of ribosome-free r-proteins in metabolically recovering HCT116 cells. **f** NR induces cell death in pre-rRNA-depleted HCT116 and A375 cells. Representative microscopy images were taken after 24 h of NR. Scale bar, 100 µm. **g** Percentage of cell death of metabolically recovering cells as determined using trypan blue exclusion assays. Data represent mean ± SD of $n = 3$ biological replicates. $P$-values (left to right): HCT116, <0.0001, <0.0001, A375, <0.0001, <0.0001, statistical analysis by two-way ANOVA. **h** NR induces apoptosis in pre-rRNA-depleted cells, which is inhibited by the pan-caspase inhibitor zVAD-FMK (1 µM). **i** Pre-rRNA synthesis resumes after CX-5461 (10 µM)/BMH-21 (1 µM) drug washout and NR. Data represent mean ± SD of $n = 3$ independent images. $P$ values (left to right): 0.077, 0.0422, statistical analysis by two-way ANOVA. Scale bar, 10 µm. For **a**, **b**, **g**, **i**, ns, not significant; ∗$p < 0.05$; ∗∗$p < 0.01$; ∗∗∗$p < 0.001$; ∗∗∗∗$p < 0.0001$.

PARP indicated that NR-mediated cell death was apoptosis, detection of which was abrogated by the addition of the pan-caspase inhibitor zVAD-FMK (Fig. 3h). Furthermore, we observed that pre-rRNA-depleted cells resumed pre-rRNA synthesis upon NR (Fig. 3i), indicating that the effects of CX-5461 and BMH-21 are reversible. Taken together, these results show that failure to accumulate pre-rRNAs under ND leads to perturbed ribosomal subunit assembly and apoptosis during NR.

**Nucleolar surveillance drives NR-mediated apoptosis.** Disruptions in ribosome assembly cause unassembled uL5/uL18 to accumulate and bind to MDM2 in an inhibitory manner, leading to the stabilization of p53 and activation of p53-dependent apoptosis[22–25,35–37]. Known as the nucleolar surveillance pathway, we hypothesized that this mode cell death was the primary mechanism by which NR induces apoptosis in pre-rRNA-depleted cells. In line with this, pre-rRNA-depleted (ND + CX-5461/ND + BMH-21) cells subjected to NR accumulated high levels of p53 whereas this was not observed in starved vehicle (ND + Vehicle) cells (Fig. 4a, Supplementary Fig. 3b). Notably, p53 was not stabilized under ND + CX-5461 or ND + BMH-21, likely due to reduced uL5/uL18 translation (Supplementary Fig. 3c). We next immunoprecipitated MDM2 to assess the expression of associated uL5/uL18. MDM2 was undetectable in control ND cells, but was upregulated in pre-rRNA-depleted cells exposed to NR (Fig. 4b). This upregulation is due to MDM2 being a downstream transcriptional target of p53. Importantly, in pre-rRNA-depleted cells, precipitation of MDM2 showed high levels of co-immunoprecipitated uL5/uL18 (Fig. 4b).

We next tested whether compromising the uL5/uL18-dependent nucleolar surveillance pathway could alleviate NR-mediated apoptosis. We therefore performed a double knockdown of uL5 and uL18 using two independent sets of siRNAs and assessed the activation of p53 and apoptotic markers. Dual depletion of uL5 and uL18 completely blocked the stabilization of p53 and greatly reduced PARP and Caspase 3 cleavage (Fig. 4c). Likewise, dual knockdown of uL5 and uL18 also decreased NR-mediated activation of p53 in A375 and A549 cells (Supplementary Fig. 3d). Independently silencing uL5 or uL18 alone also strongly inhibited p53 (Supplementary Fig. 3e). Further support for the role of p53 in NR-mediated apoptosis was obtained using p53 wild-type or knockout HCT116 cells. Pre-rRNA-depleted HCT116 p53 −/− cells exposed to NR exhibited lower levels of apoptosis compared to wild-type counterparts as shown by decreased activation of apoptotic markers and Annexin V staining (Fig. 4d, e). Finally, siRNAs targeting p53 or PUMA (the downstream effector of p53) also strongly alleviated NR-induced apoptosis in HCT116 and A375 cells (Supplementary Figs. 4a–c, 5a).

Taken together, these data show that cancer cells accumulate unprocessed rRNAs under starvation in order to escape p53-mediated apoptosis during nutrient refeeding (Fig. 4f). Pre-rRNAs that accumulate during starvation are utilized for ribosome biogenesis upon metabolic recovery (see also pulse-chase labeling Fig. 2a, b). Failure to accumulate pre-rRNAs under starvation results in unassembled uL5/uL18 binding to MDM2, leading to nutrient-induced p53-mediated apoptosis (Fig. 4f).

**Glutamine metabolically activates p53 by inducing uL5/uL18 translation.** The activation of p53 through the nucleolar surveillance pathway requires ongoing de novo protein synthesis of uL5 and uL18[35,38]. Thus, the accumulation of p53 serves as a proxy marker for active uL5/uL18 mRNA translation. Given this, we found that restoring glutamine alone was sufficient to rescue p53 in pre-rRNA-depleted HCT116 and A375 cells (Fig. 5a, Supplementary Fig. 5b). In contrast, restoration of leucine,

tyrosine, and methionine had no effect on p53 (Supplementary Fig. 5c). In addition, dual knockdown of uL5/uL18 inhibited the activation of p53 by glutamine (Supplementary Fig. 5d). These results indicate that glutamine metabolically activates p53 in pre-rRNA-depleted cells by inducing uL5/uL18 protein synthesis. Finally, glutamine restoration also reactivated mTORC1, as shown by the increase of S6K (Thr 389) phosphorylation, a known mTORC1 substrate (Fig. 5a). This finding is consistent with the role of glutamine in activating mTORC1[39,40]. This prompted us to examine whether mTORC1 is required for the metabolic stabilization of p53. To test this, pre-rRNA-depleted cells were re-fed glutamine in the presence of rapamycin or Torin1, two highly potent and specific mTORC1 inhibitors. However, rapamycin or Torin1 had no inhibitory effect on p53 (Fig. 5b). We therefore wondered whether other pathways that regulate protein synthesis, such as the Ras/Raf/MEK/ERK signaling cascade, might regulate glutamine-mediated synthesis of uL5/uL18. In support of this, glutamine restoration rescued MEK and ERK phosphorylation in pre-rRNA-depleted A375 cells (Fig. 5c). To test whether the metabolic stabilization of p53 is dependent on Raf/MEK/ERK signaling, pre-rRNA-depleted cells were re-fed glutamine in combination with a pan-Raf inhibitor, LY3009210 (400 nM) or AZ628 (500 nM). Strikingly, both pan-Raf inhibitors blocked the activation of p53 by glutamine (Fig. 5d). As expected, LY3009210 or AZ628 treatment strongly decreased ERK and MEK phosphorylation (Fig. 5d).

We next sought to understand the potential regulatory effects of ERK signaling on uL5/uL18 translation. Given that mTORC1 primarily regulates translation initiation[41], and that mTORC1 suppression did not inhibit p53 (Fig. 5b), we reasoned that ERK might regulate uL5/uL18 via the elongation step of translation. Indeed, in unstressed cells, ERK facilitates mRNA translation by inhibiting eukaryotic elongation factor-2 kinase (eEF2K), a key negative regulator of protein synthesis (Fig. 5e)[42,43]. Upon eEF2K inhibition, its downstream target eukaryotic elongation factor-2 (eEF2) remains in its active unphosphorylated form to mediate global translation elongation (Fig. 5e)[42,43]. Thus, we hypothesized that uL5/uL18 translation is dependent on ERK-mediated activation of eEF2 (Fig. 5e). We therefore assayed levels of eEF2 phosphorylation; phosphorylated eEF2 (Thr56) being the inactivated form that leads to arrest of general protein synthesis[42,43]. Inhibiting Raf with LY3009210 or AZ628 increased eEF2 phosphorylation in both starved control and pre-rRNA-depleted cells (Fig. 5d). We also tested if inhibiting the downstream kinases MEK and ERK would produce the same effect on eEF2 and p53. Pharmacological inhibition of MEK (U0126, PD98059) or ERK (LY3214996, GDC-0994) similarly induced eEF2 phosphorylation and p53 inhibition (Fig. 5f). Likewise, in A375 cells, inhibition of Raf (LY3009210), MEK (U0126), and ERK (LY3214996) completely blocked p53 expression (Fig. 5g). Next, to assess the effect of Raf inhibition on uL5/uL18 mRNA translation, we measured the expression of translationally active polysome-bound mRNAs (see "Methods"). ND alone decreased the proportion of polysome-bound uL5/uL18 mRNAs (Fig. 5h). Importantly, glutamine restoration rescued uL5/uL18 mRNA translation, but was inhibited by co-incubation with LY3009210 (Fig. 5h). Consistent with the role of eEF2 in general mRNA translation[42,43], a similar effect was observed with uL23 and S3 polysome-bound mRNAs (Fig. 5h).

**Glutamine deprivation confers cancer cells resistance to RNA Pol I inhibitors by blocking the nucleolar surveillance pathway.** Since glutamine restoration stabilized p53 in pre-rRNA-depleted cells (Fig. 5a), we next tested the effect of glutamine deprivation on the activation of p53 by CX-5461. Strikingly, glutamine

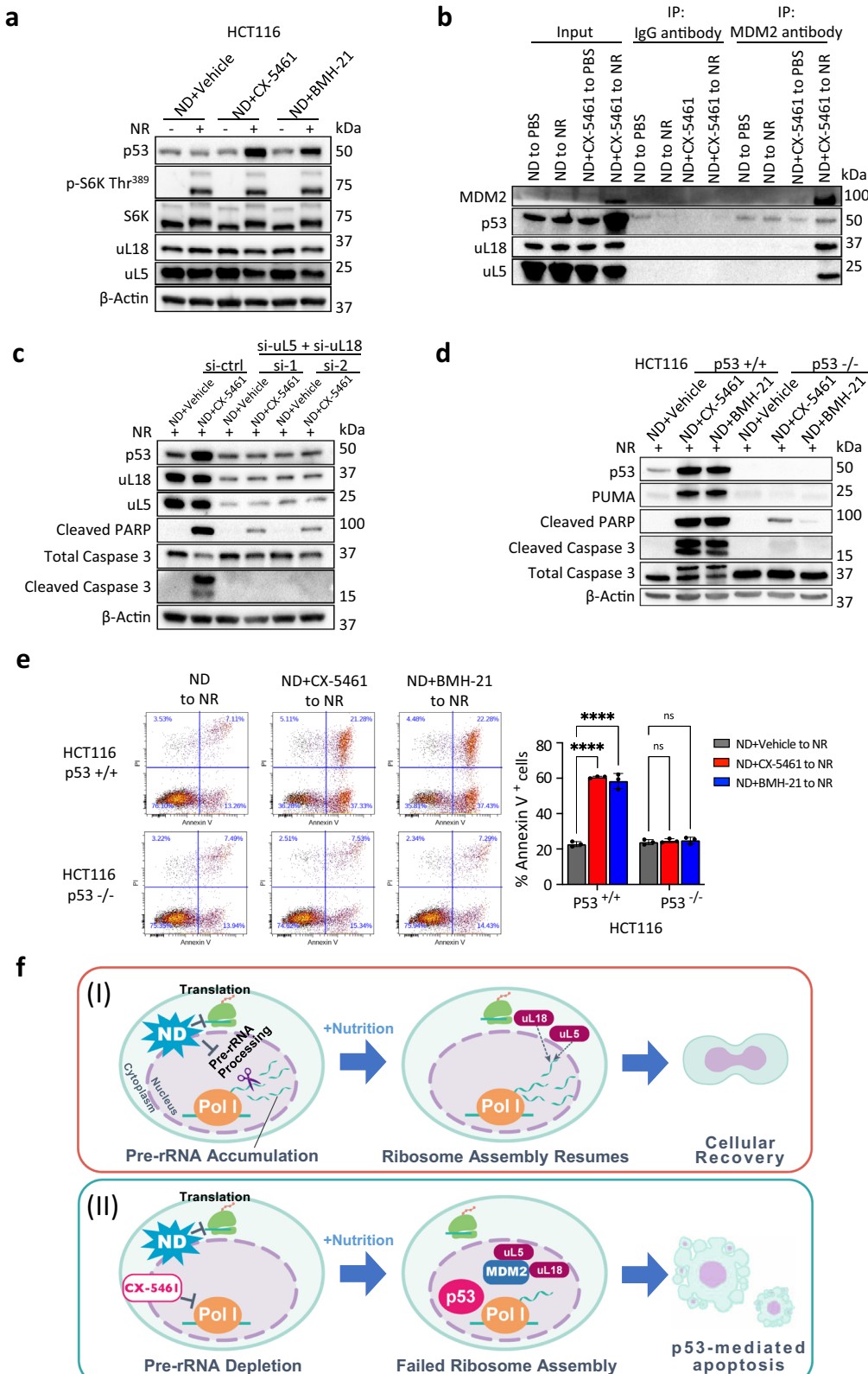

deprivation blocked the activation of p53 by 1 μM CX-5461, while the withdrawal of glucose or other amino acids had no effect (Fig. 6a, Supplementary Fig. 6a). Glutamine deprivation did not inhibit p53 stabilization by the MDM2 inhibitor AMG 232 (Supplementary Fig. 6b). Suppression of p53 transcriptional activity under glutamine deprivation was confirmed by measuring p21 mRNA (Supplementary Fig. 6c). The inhibition of p53 was

not due to reduced uL5/uL18 translation, as glutamine deprivation did not noticeably inhibit uL5/uL18 nascent protein synthesis (Supplementary Fig. 6d). Interestingly, 1 μM CX-5461 failed to decrease pre-rRNA expression under glutamine deprivation (Supplementary Fig. 6e), however, increasing the dose to 10 μM rescued pre-rRNA inhibition and p53 activation (Supplementary Fig. 6f, g). CX-5461 caused uL5/uL18 to co-immunoprecipitate

**Fig. 4 Nucleolar surveillance drives NR-mediated apoptosis. a** NR induces p53 in pre-rRNA depleted HCT116 cells as assessed using western blotting. **b** Co-immunoprecipitation of p53, uL5, and uL18 with MDM2. **c** Dual knockdown of uL5 and uL18 by siRNA inhibits NR-mediated activation of p53 and apoptotic markers. **d** NR induces apoptosis in HCT116 p53+/+ cells but not p53−/− cells. **e** NR induces apoptosis in pre-rRNA depleted HCT116 p53+/+ cells but not HCT116 p53−/− cells, as determined using Annexin V-PI flow cytometry. Data represent mean ± SD of $n = 3$ technical replicates. $P$ values (left to right): p53+/+, $7.6 \times 10^{-7}$, $7.9 \times 10^{-5}$, p53−/−, <0.0001, <0.0001 (ns, not significant; ∗∗∗∗$p < 0.0001$, statistical analysis by two-way ANOVA). **f** (I) pre-rRNA intermediates accumulate in nutrient deprived cancer cells. In parallel, ND suppresses global mRNA translation. Following nutrient addition (NR), accumulated pre-rRNAs are bound by newly translated uL5 and uL18 for ribosome assembly. (II) Treatment with CX-5461 (1 μM) prevents cancer cells from accumulating pre-rRNAs under ND. Following NR, upon the absence of endogenous pre-rRNAs, uL5 and uL18 binds to MDM2, leading to p53 stabilization and nutrient-induced apoptosis.

with MDM2 under control conditions but not under glutamine deprivation (Supplementary Fig. 6h). Glutamine deprivation inhibited the activation of p53 by BMH-21 and two other Pol I inhibitors, CX-3543 and actinomycin D (Fig. 6b). Glutamine deprivation also inhibited p53 in A375, A549, U2OS, LNCaP, and MKN45 cells (Fig. 6c), and conferred resistance to CX-5461 in HCT116 cells (Fig. 6d) as well as LNCaP and MKN45 cells (Supplementary Fig. 7a). Glucose deprivation, which does not inhibit p53, did not confer resistance to CX-5461 (Fig. 6d). Overexpression of p53 under glutamine deprivation rescued the growth suppressive effects of CX-5461 (Supplementary Fig. 7b). In addition, rescuing p53 with the non-genotoxic MDM2 inhibitor AMG 232 also inhibited the viability of glutamine-deprived cells (Supplementary Fig. 7c).

Solid tumors contain low levels of glutamine due to rapid consumption by tumor cells[34]. Thus, we predicted that CX-5461 would fail to stabilize p53 in vivo. Indeed, p53 was suppressed under 0.4 mM glutamine (Fig. 6e), which is approximately the concentration of glutamine found in tumor xenografts[34]. We selected HCT116 as our model cell line, since CX-5461 has been shown to potently inhibit HCT116 in vitro cell proliferation at an IC50 167 nM[30]. Oral gavage of CX-5461 at the highest long-term tolerable dose (50 mg/kg) failed to significantly inhibit tumor growth (Fig. 6f). We used AMG 232 as a positive control for p53-mediated tumor suppression. Oral gavage of AMG 232 (50 mg/kg) at the same dose and schedule as CX-5461 significantly suppressed HCT116 tumor growth (Fig. 6f). Furthermore, IP injection of CX-5461 up to 200 mg/kg to HCT116 tumors failed to activate p53 while AMG 232 (50 mg/kg) robustly induced p53 (Fig. 6g). Similarly, AMG 232 but not CX-5461 stabilized p53 in A375 tumors (Fig. 6h). These results indicate that tumoral glutamine deficiency hampers the antitumor activity of Pol I inhibitors by blocking the p53-dependent nucleolar surveillance pathway.

## Discussion
Unstable vasculature in solid tumors leads to the development of tissue microenvironments that fluctuate in nutrient availability[7]. Tumor cells residing in such environments adapt to nutrient depletion by suppressing processes that heavily utilize ATP such as ribosome biogenesis[10–12], but how cells recover during nutrient restoration remains to be elucidated. Here, we report that metabolically stressed mammalian cells accumulate unprocessed rRNAs in order to escape nucleolar surveillance upon nutrient stress termination. We first observed that ND inhibits pre-rRNA processing, leading to the accumulation of pre-rRNA intermediates in nutrient starved cells. Upon nutrient restoration, recovering cells resume ribosome biogenesis and pre-rRNAs accumulated during stress are converted into mature rRNAs. Cells that fail to accumulate pre-rRNAs under ND undergo cell-lethal p53 activation during nutrient refeeding. Mechanistically, p53 is activated by unassembled uL5/uL18 binding and inhibiting MDM2. Our data shows that pre-rRNA accumulation during nutrient deprivation is a stress adaptation that restrains the

MDM2-binding activity of uL5 and uL18 during fluctuations in nutrient availability.

During the preparation of this manuscript, CX-5461 was reported to possess additional inhibitory activities against Topoisomerase II[44,45]. This presents a limitation to our manuscript since we employed CX-5461 to inhibit pre-rRNA accumulation under ND. We addressed this limitation by employing BMH-21 alongside CX-5461 in our experiments.

We further identified that the activation of p53 through the uL5/uL18-dependent nucleolar surveillance pathway is dependent on exogenous glutamine. In pre-starved cells depleted of pre-rRNAs, refeeding glutamine was sufficient to stabilize p53, compatible with rescue of uL5/uL18 translation. Surprisingly, despite the known importance of mTORC1 in the translational control of 5′TOP mRNAs (that encode ribosomal proteins including uL5 and uL18)[46,47], rapamycin or Torin1 treatment did not affect the stabilization of p53 by glutamine. Since the accumulation of p53 protein is a marker of uL5 and uL18 synthesis[35,38], this observation suggested that uL5/uL18 translation was not solely dependent on mTORC1. Rather, we found that pharmacologically inhibiting Raf/MEK/ERK completely blocked the accumulation of p53. Raf/MEK/ERK inhibition appeared to arrest the synthesis of uL5 and uL18 by inactivating eEF2. Altogether, these results suggest that glutamine-induced synthesis of uL5 and uL18 is facilitated by the Ras/Raf/MEK/ERK-eEF2K-eEF2 axis.

Finally, we found that glutamine deprivation confers resistance to CX-5461 by blocking the stabilization of p53. This finding was supported in vivo where CX-5461 failed to stabilize p53, which could explain the little inhibitor effect on HCT116 tumor growth, particularly given the reported low glutamine tumor microenvironment[34]. In line with our findings, another study observed that p53 wild-type HCT116 tumors are highly resistant to CX-5461[48]. This is despite another study demonstrating that HCT116 cells are remarkably sensitive to CX-5461 in vitro[30]. Thus, these results indicate that CX-5461 may display limited therapeutic benefit to solid tumors due to tumoral glutamine deficiency. Notably, glutamine deprivation did not decrease uL5/uL18 protein synthesis, indicating that the mechanism of p53 inhibition is independent of uL5/uL18 mRNA translation. Altogether, our data shows that tumoral glutamine deficiency hampers the effectiveness of Pol I inhibitors by blocking the nucleolar surveillance pathway. These findings offer an explanation as to why the efficacy of CX-5461 is not associated with p53 wild-type status in solid tumor types. Further, these results may explain why CX-5461 is highly effective against blood cancers[25,27–29], in which severe pathological nutrient deprivation does not occur. Indeed, plasma glutamine concentrations are approximately 0.8 mM[49], which is twice higher than the concentration at which p53 was found to be inhibited in this study (Fig. 4e). Further, since the rate of ribosome biogenesis is dependent on nutrient availability, this implies that cells of hematological malignancies possess higher rates of ribosome biogenesis than cells of solid tumors and are therefore more vulnerable to Pol I inhibitors.

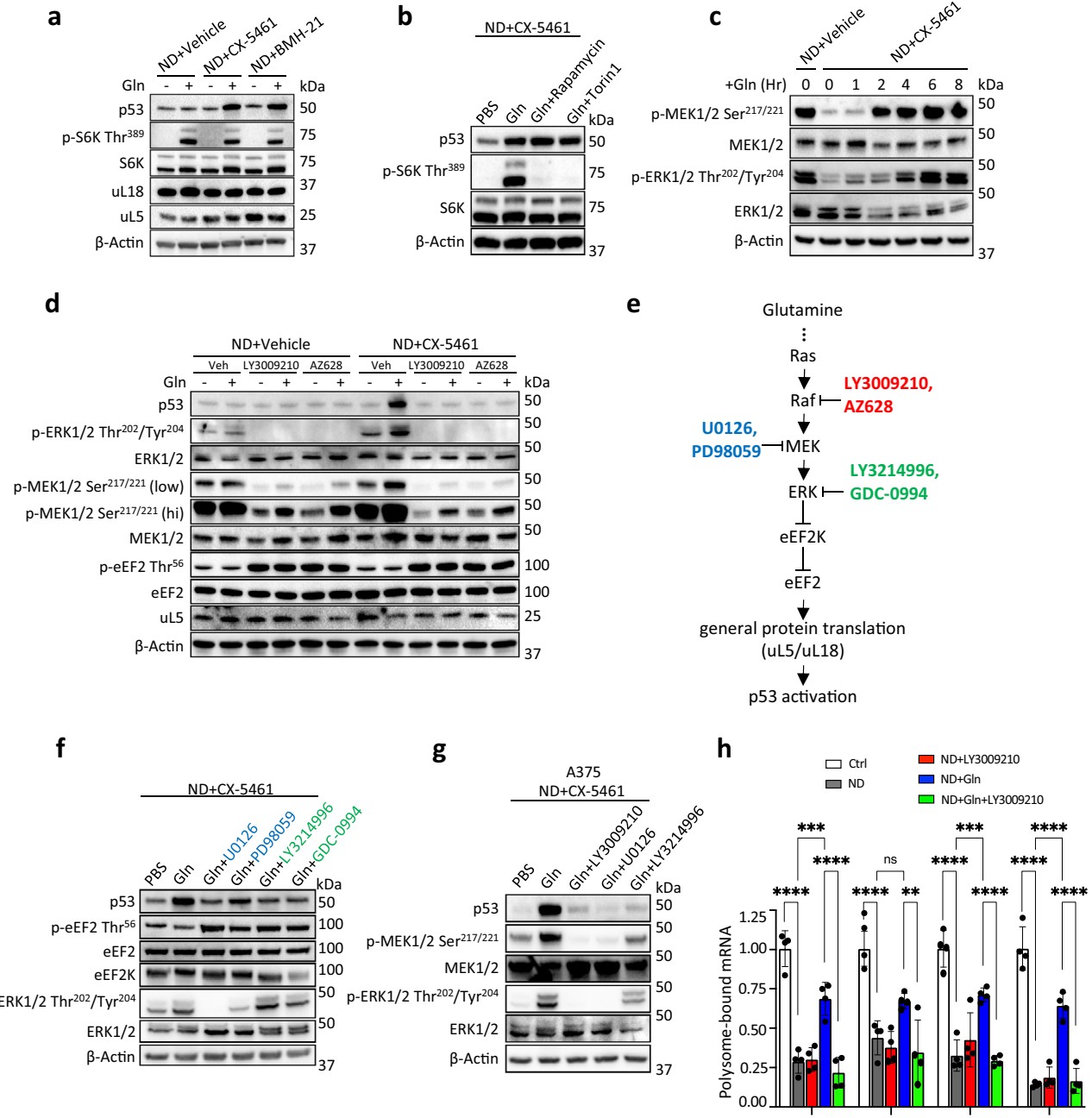

**Fig. 5 Glutamine metabolically activates p53 by inducing uL5/uL18 translation. a** Glutamine (gln) activates p53 in pre-rRNA depleted HCT116 cells as assessed by western blotting. HCT116 were cultured in ND medium with vehicle (NaH$_2$PO$_4$), CX-5461 (10 μM), or BMH-21 (1 μM) for 24 h. After 24 h the old treatment media was replaced with ND media containing PBS or Gln (4 mM) for 8 h. **b** Inhibiting mTORC1 with Rapamycin (1 μM) or Torin1 (1 μM) does not decrease the stabilization of p53 by glutamine. **c** Glutamine restoration rescues MEK and ERK phosphorylation in pre-rRNA-depleted A375 cells. **d** Inhibiting Raf with LY3009210 (400 nM) or AZ628 (500 nM) inhibits the metabolic activation of p53 by glutamine (gln). **e** Schematic: Glutamine restoration activates Ras/Raf/MEK/ERK signaling, which inhibits eEF2K. Upon eEF2K inhibition, eEF2 remains in its active form to facilitate global protein synthesis (including uL5/uL18 translation). Pharmacological inhibition of Raf/MEK/ERK leads to eEF2k activation, which in turn phosphorylates and inactivates eEF2. **f** Inhibiting MEK (10 μM U0126, 50 μM PD98059) or ERK (1 μM LY3214996, 0.1 μM GDC-0994) inhibits the metabolic activation of p53 by glutamine (gln). **g** Inhibiting Raf (400 nM LY3009210), MEK (10 μM U0126) and ERK (1 μM LY3214996) inhibits the metabolic activation of p53 by glutamine (gln) in A375 cells. **h** Expression of polysome-bound r-protein mRNAs. HCT116 cells were placed under ND for 24 h. After 24 h the old treatment media was replaced with ND media containing 4 mM Gln, LY3009210 (400 nM), or Gln + LY3009210. Polysomes were collected via ultracentrifugation on a 40% sucrose cushion and total RNA was extracted and assayed using qRT-PCR for uL5, uL18, uL23, and S3. Data represent mean ± SD of $n = 4$ technical replicates. *P*-values (left to right): uL5, <0.0001, 0.0003, <0.0001, uL18, <0.0001, 0.0052, uL23, <0.0001, 0.0003, <0.0001, S3, <0.0001, <0.0001, <0.0001 (∗∗*p* < 0.01; ∗∗∗*p* < 0.001; ∗∗∗∗*p* < 0.0001, statistical analysis by two-way ANOVA).

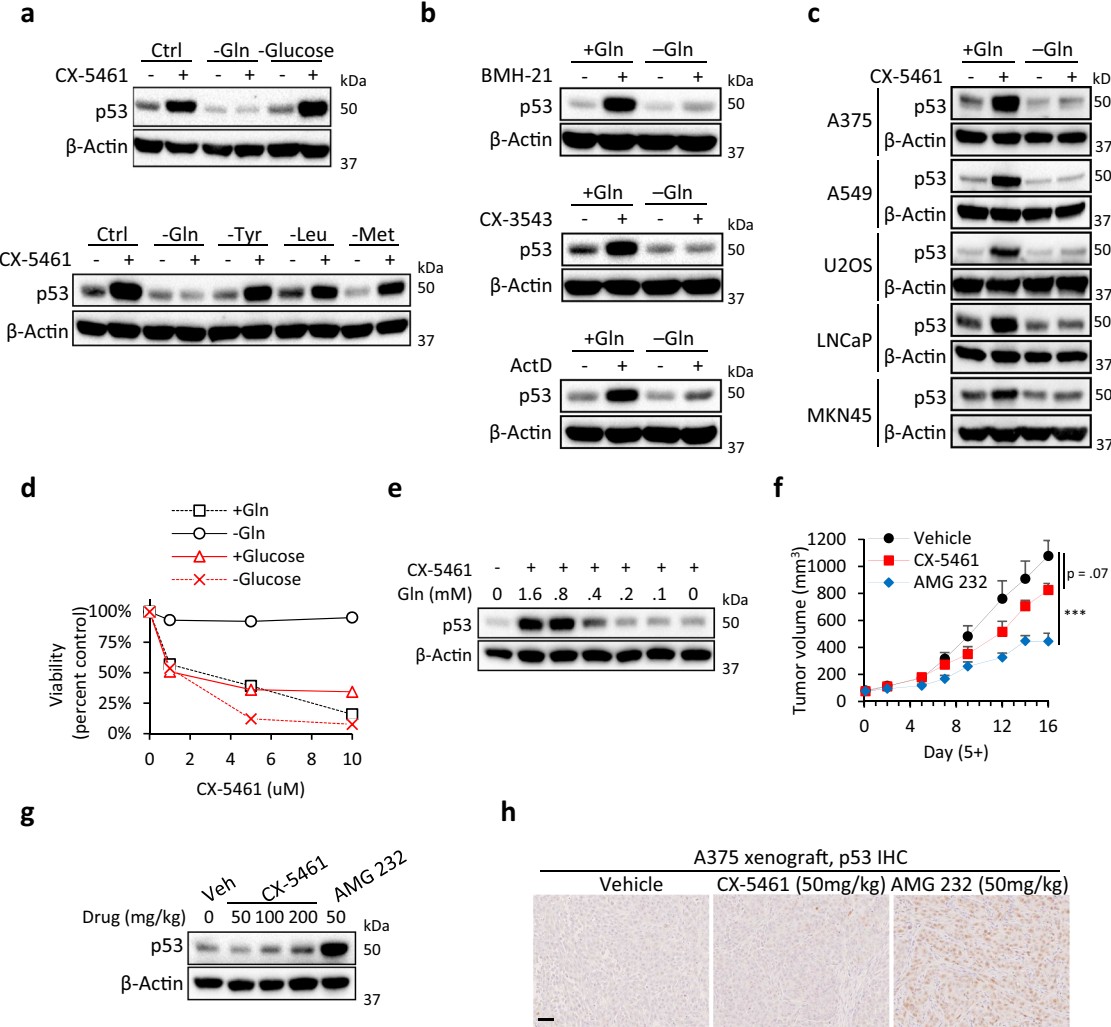

**Fig. 6 Glutamine deprivation confers cancer cells resistance to RNA Pol I inhibitors by blocking the nucleolar surveillance pathway. a** Glutamine deprivation inhibits the activation of p53 by CX-5461 treatment. HCT116 cells were cultured in the respective starvation medium for 16 h followed by CX-5461 (1 μM) for 8 h. **b** Glutamine deprivation inhibits p53 activation by BMH-21 (1 μM), CX-3543 (1 μM), and ActD as assessed by western blotting. **c** Glutamine deprivation inhibits p53 activation by CX-5461 (1 μM) in A375, A549, U2OS, LNCaP, and MKN45 cells. **d** Deprivation of glutamine but not glucose conferred resistance to CX-5461 (1 μM) as assessed using MTT assay. Data points of each treatment were normalized to the respective vehicle-treated metabolic control. Data represent mean ± SD of $n = 6$ biological replicates. **e** p53 activation by CX-5461 was suppressed under 0.4 mM glutamine as assessed by western blotting. **f** CX-5461 failed to inhibit HCT116 tumor growth. Mice were orally dosed with vehicle (NaH2PO4), CX-5461 (50 mg/kg), or AMG 232 (50 mg/kg) once daily three times per week. Drug treatment started 5 days post implantation on tumors of approximately 75 mm$^3$. Data represent mean ± SEM of $n = 8$ mice/group. P values: Vehicle/CX-5461, 0.078, Vehicle/AMG 232, $2.4 \times 10^{-4}$ (*$p < 0.05$; **$p < 0.01$; ***$p < 0.001$). **g** CX-5461 does not acutely stabilize p53 in HCT116 tumors. Mice were IP injected with vehicle (NaH2PO4), CX-5461 (50, 100, 200 mg/kg), or AMG 232 (50 mg/kg) and tumors were harvested after 8 h for western blot analysis. (h) p53 IHC staining in A375 tumors following vehicle or drug IP administration. Scale bar represents 50 μm.

In summary, our study unveils a stress-adaptive program that suppresses p53 during fluctuations in nutrient availability. Accumulation of unprocessed rRNAs emerges as a critical survival mechanism that is exploited by starving tumor cells to complete metabolic recovery during nutrient restoration. Further, solid tumors do not accumulate p53 in response to Pol I inhibitors, which could be explained to be inherently depleted of glutamine, highlighting how the metabolic landscape of the tumor microenvironment can deeply influence the therapeutic response to nucleolar function disruptors.

## Methods

**Cell lines, Reagents, and siRNA transfections.** HCT116, A375, A549, U2OS, and MKN45, cells were purchased from the American Type Culture Collection (Manassas, VA, USA). HCT116 p53+/+ and p53−/− isogenic human colon

cancer cells were kindly provided by Bert Vogelstein (Johns Hopkins University). HCT116, A375, A549, and U2OS cells were grown in Dulbecco's modified Eagle's medium (DMEM) (Nacalai Tesque, Kyoto, Japan), supplemented with 10% fetal bovine serum (FBS) (Thermo Fisher Scientific). MKN45 cells were maintained in RPMI (Nacalai Tesque, Kyoto, Japan), supplemented with 10% FBS. Cells were maintained at 37 °C in a 5% $CO_2$ atmosphere in a humidified incubator. Nutrient deprivation medium was prepared to contain inorganic salts, i.e., 0.2 g/l $CaCl_2$ (anhydrous), 0.1 mg/l Fe(NO$_3$)$_3$/9H$_2$O, 0.4 g/l KCl, 97.67 mg/l MgSO$_4$ (anhydrous), 6.4 g/l NaCl, 3.7 g/l NaHCO$_3$, 0.125 g/l NaH$_2$PO$_4$/H$_2$O, and 15 mg/l Phenol Red, according to the composition of DMEM. Glutamine deprivation was performed with DMEM without glutamine (Fujifilm Wako 045-30285) supplemented with 10% FBS. Glutamine replete medium was made by adding 4 mM L-glutamine (Fujifilm Wako) to the glutamine deprivation medium. Glucose deprivation was performed with DMEM without glucose (Fujifilm Wako 042-32255) supplemented with 10% FBS. Glucose replete medium was made by adding 25 mM D-glucose (Fujifilm Wako) to the glucose deprivation medium. Tyrosine, leucine, and methionine medium were prepared according to the composition of DMEM (Nissui, Tokyo, Japan).

The compounds used in this study were obtained from: CX-5461 (MedChemExpress), AMG 232 (MedChemExpress), BMH-21 (Selleck), Camptothecin (Fujifilm Wako), Rapamycin (Fujifilm Wako), Torin1 (Fujifilm Wako), BEZ235 (Funakoshi), 5-Fluorouridine (TCI chemicals), Actinomycin D (Fujifilm Wako), zVAD-FMK (Fujifilm Wako), LY3009120 (Selleck), AZ628 (Selleck), U0126 (Selleck), PD98059 (Selleck), LY3214996 (Selleck), GDC-0994 (Selleck).

Cancer cells were transfected at ~30% confluency in 6-well plates with 5 nM control or target siRNA using Lipofectamine RNAiMAX (Invitrogen) according to the instructions of the manufacturer. All siRNAs were purchased from Thermo Fisher Scientific: negative control (4390843), p53 (s607, s605), uL18 (RPL5) (s56731, s12152), uL5 (RPL11) (s533180, s533179), PUMA (s25842, s25840).

**Trypan Blue assays, MTT assays**. For trypan blue exclusion assays, $6 \times 10^5$ cells (HCT116, A375, A549, U2OS) were seeded in 6-well plates and the percentage of cell death was determined after 24 h ND. Cell counting was performed with the TC20 Automated Cell Counter (Bio-Rad) according to the instructions of the manufacturer. After 24 h ND, the treatment medium was set aside, adherent cells were detached with 100 μL trypsin and suspended with 900 μL of the treatment media. One hundred microliters of the suspension was mixed with 100 μL trypan blue solution (Bio-Rad) and analyzed with the TC20 Cell Counter.

For MTT assays, $6–10 \times 10^5$ cancer cells were seeded in six-well plates, and after treatment, the cell media was aspirated and replaced with medium containing MTT (0.5 mg/mL) (Fujifilm Wako) for 3 h and lysed with DMSO. Absorbance was measured at 570 nm with a plate reader.

**Western blotting**. Cells were lysed with RIPA buffer containing a protease inhibitor cocktail (P8340 Sigma), phosphatase inhibitors (P0044 and P5726 Sigma), and 1 mM PMSF. Protein quantification was performed using BCA kit (Pierce). Cell lysates were applied to a 10% polyacrylamide gel and transferred to a nitrocellulose membrane (Thermo Fisher Scientific). The membrane was incubated with antibodies that target p53 (1:1000, Calbiochem Ab-6 clone DO-1), b-actin (1:1000, Sigma-Aldrich), phospho-S6K (T389) (1:1000, Cell Signaling Technology), S6K (1:1000, Cell Signaling Technology), phospho-S6 (S235/236) (1:1000, Cell Signaling Technology #2211), S6 (1:1000, Cell Signaling Technology), phospho-4E-BP1 (Thr37/46) (1:1000, Cell Signaling Technology), 4E-BP1 (1:1000, Cell Signaling Technology), HIF-1α (1:1000 Novus Biologicals), Cleaved PARP (1:1000, Cell Signaling Technology), Caspase 3 (1:1000, Cell Signaling Technology), Cleaved Caspase 3 (1:1000, Cell Signaling Technology), PUMA (1:1000, Cell Signaling Technology), RPL5 (1:1000, kindly provided by Dr. Siniša Volarević), RPL11 (1:1000, kindly provided by Dr. Siniša Volarević), phospho-MEK1/2 (Ser217/221) (1:500 Cell Signaling Technology), MEK1/2 (1:500 Cell Signaling Technology), phospho-ERK1/2 (Thr202/Tyr204) (1:500 Cell Signaling Technology), ERK1/2 (Thr202/Tyr204) (1:500 Cell Signaling Technology), phospho-eEF2 (Thr56) (1:1000 Cell Signaling Technology), eEF2 (1:1000 Cell Signaling Technology), eEF2K (1:1000 Cell Signaling Technology), followed by incubation with horseradish peroxidase-conjugated secondary antibodies (1:5000, Sigma-Aldrich). Signals were detected using enhanced chemiluminescence detection reagents (Thermo Fisher Scientific) and images were acquired using a luminescent image analyzer (LAS3000, Fuji-Film, Japan).

The Click-IT™ AHA (L-Azidohomoalanine) kit (Thermofisher Scientific, Catalog #C10102) was used to assess nascent protein production. AHA labeling, biotinylation, and affinity purification was performed according to the instructions of the manufacturer.

**Isolation of ribosome-free fractions and polysome-bound mRNA**. Metabolically recovering (ND to NR) HCT116 cells $(4 \times 10^8)$ were washed in PBS containing 100 μg/mL cycloheximide twice. Total cell extracts were suspended in 2.5 mL polysome buffer (20 mM Tris, 10 Mm MgCl$_2$, 300 mM KCl, 10 mM dithiothreitol, 100 units/mL RNasin, 100 μg/mL cycloheximide, protease and phosphatase inhibitors, pH 7.4) and gently sheared 4× using a 26-gauge needle. Total cell extracts were centrifuged ($20,000 \times g$ 30 min) to pellet cell debris. The resulting cell extracts were layered over a 20% sucrose cushion (polysome buffer + 20% w/v sucrose) and ultracentrifuged (Beckman Coulter TLA100.3 Fixed-Angle Rotor, $149,000 \times g$ 2 h). The supernatant containing the non-ribosomal fraction was later assessed using western blotting. To isolate polysome-bound mRNAs, cell extracts were layered over a 40% sucrose cushion (polysome buffer + 40% w/v sucrose) and the ribosomal pellet was collected for RNA extraction and qRT-PCR[35].

**Northern Blotting**. Steady-state pre-rRNA analysis was performed by northern blotting[19]. Total RNA (5 μg) was separated by electrophoresis on agarose denaturing gels (6% formaldehyde/1.2% agarose in 50 mM HEPES, 1 mM EDTA). At the end of the migration, the gels were processed for transfer by capillary overnight in 10× saline sodium citrate buffer onto nylon membranes (GE Healthcare). The membranes were prehybridized for 1 h at 65C in 50% formamide, 5× SSPE (5 mM EDTA, 50 mM NaH$_2$PO$_4$, 750 mM NaCl), 5× Denhardt's solution (0.1% Ficoll 400, 0.1% polyvinyl pyrrolidone, 0.1% bovine serum albumin fraction V), 1% SDS, 200 μg/ml fish sperm DNA solution (Roche). Antisense oligonucleotide probes labeled with [g-$^{32}$P]-ATP with T4 polynucleotide kinase (New England

Biolabs) were added and incubated for 1 h at 65C and then overnight at 37C. The membranes were washed three times for 3 min in 2× SSC and exposed to Fuji imaging screen (Fujifilm). After 24 h to 72 h exposure, the signal was acquired with a Phosphorimager (FLA-7000; Fujifilm) and quantitated with the native Multi-Gauge Software (Fujifilm, v 3.1). Membranes were stripped in boiling 0.1% SDS in between successive hybridizations.

Sequences of oligonucleotides used as probes:
LD1844 (human 5′ETS):
CGGAGGCCCAACCTCTCCGACGACAGGTCGCCAGAGGACAGCGTGT-CAGC
LD2122 (human ITS1):
GCCCTCCGGGCTCCGTTAATGATC
LD1828 (human ITS2):
CTGCGAGGGAACCCCCAGCCGCGCA
LD4096 (mouse 5′ETS):
AGAGAAAAGAGCGGAGGTTCGGGACTCCAA
LD4099 (mouse ITS1):
ACGCCGCCGCTCCTCCACAGTCTCCCGTT
LD4100 (mouse ITS2):
ACTGGTGAGGCAGCGGTCCGGGAGGCGCCGACG.

**RNA Isolation and quantitative RT-PCR**. Total RNA was extracted from cells using the Isogen reagent (Nippon Gene, Toyama, Japan), converted to cDNA by using the Prime Script reverse transcriptase (Takara, Shiga, Japan) as per the manufacturer's instructions, and used for quantitative real-time PCR amplification using SYBR Green (Takara). Target RNA expression was normalized to β2m mRNA. Quantification of the pre-rRNA half-life was performed according to a previous report[50]. Briefly, cells were treated to an excess concentration of actinomycin D (20 μM) that blocks transcription within 30 s[50], and pre-rRNA expression was tracked over 1 h.

The following primer sequences were used:
For pre-rRNA expression:
5′-CCGCGCTCTACCTTACCTACCT-3′(forward);
5′-GCATGGCTTAATCTTTGAGACAAG-3′(reverse).
For human p21:
5′-TGTCCGTCAGAACCCATGC-3′;
5′-AAAGTCGAAGTTCCATCGCTC-3′.
For human β2m:
5′-AGATGAGTATGCCTGCCGTG-3′;
5′-CATCCAATCCAAATGCGGCA-3′.
For uL5 (RPL11):
5′-AAAGGTGCGGGAGTATGAGTT-3′;
5′-TCCAGGCCGTAGATACCAATG-3′.
For uL18 (RPL5):
5′-GCTCGGAAACGCTTGGTGATA-3′;
5′-CCCTCTATACGGGCATAAGCAAT-3′.
For uL23 (RPL23):
5′-TCCTCTGGTGCGAAATTCCG-3′;
5′-CGTCCCTTGATCCCCTTCAC-3′.
For S3 (RPS3):
5′-AGAGGAAGTTTGTCGCTGATG-3′;
5′-GCACCTCAACTCCAGAGTAGC-3′.

**Annexin V-PI flow cytometry**. After treatment, cancer cells were trypsinized, centrifuged ($500 \times g$, 5 min) and resuspended in 1X Binding buffer with Annexin V and propidium iodide (PI) with a Annexin V-FITC Apoptosis Detection Kit (Abcam), according to the instructions of the manufacturer. Cells were analyzed on a BAD FACSverse flow cytometry instrument.

**5-Fluorouridine metabolic labeling and microscopy**. To measure pre-rRNA biosynthesis, cells were seeded at 50% confluence in six-well plates containing round cover glasses (12CIR-1D; Thermo Fisher Scientific). After treatment, cells were incubated with 10 μM 5-Fluorouridine (5FU) (TCI chemicals) for 3 h, followed by fixation with methanol at −20 °C for 30 min and blocking with milk (5% in PBS). 5FU was visualized using primary BrdU antibody (1:500 Sigma-Aldrich) and the nucleolus was visualized by staining for nucleolin (1:1000 Cell signaling technology). Detection of primary antibodies was performed with Alexa Fluor 488 Goat anti-Mouse IgG (1:2000 Thermo Fisher Scientific) and Alexa Fluor 594 Goat anti-Rabbit IgG (1:2000 Thermo Fisher Scientific). Slides were mounted with Prolong Gold Antifade Mountant with DAPI (Thermo Fisher Scientific) and viewed using BZ-X710 fluorescence microscope or LSM700 confocal microscope (CLSM, Carl Zeiss, Jena, Germany). For 5FU fluorescence quantification, three images were taken at random per treatment with a 100× objective lens and signal intensity was quantified from using ImageJ software. 5FU intensity for each image was divided by the number of cells and fold change was calculated by setting the control cells to one.

**Cell line xenograft murine model**. All animal care procedures were in accordance with institutional guidelines approved by the University of Tokyo. HCT116 cells

$(5 \times 10^6$ in 100 μL PBS) were inoculated subcutaneously in the right flank of female 8 weeks old nude mice. Established tumors ($\sim$110–120 mm³) were randomized into vehicle, CX-5461 (50 mg/kg), or AMG 232 (50 mg/kg) treatment groups. Mice were treated with either oral gavage three times per week or i.p. administration as indicated. Tumor volume was measured by external caliper, and tumor volume was calculated using the formula (l × w2)/2, where w = width and l = length in mm of the tumor. Mouse body weight was measured three times per week. After treatments, animals were euthanized, and tumors were harvested for further analysis. Dissection of tumors to peripheral and core regions were performed[34].

**Histological analysis and immunohistochemistry.** Tumor tissues were fixed in 4% paraformaldehyde (PFA), embedded in paraffin, and subjected to hematoxylin and eosin (HE), periodic acid-Schiff (PAS) staining, and p53 staining.

**Metabolite extraction and metabolome analysis.** Metabolites of cancer cells and tumor tissues were measured in each sample using CE-MS (Agilent Technologies, Santa Clara, CA). For cationic compounds, a fused silica capillary (50 mm i.d. 3 100 cm) was used with 1 M formic acid as the electrolyte. Methanol/water (50% v/v) containing 0.1 mM hexakis (2,2-difluoroethoxy) phosphazene was run as the sheath liquid at 10 mL/min. ESI-TOFMS was performed in positive ion mode, and the capillary voltage was set to 4 kV. Automatic recalibration of each acquired spectrum was obtained using the reference standards ([13 C isotopic ion of a protonated methanol dimer (2 MeOH+H)]+, $m/z$ 66.0632) and ([hexakis(2,2- difluoroethoxy) phosphazene +H]+, $m/z$ 622.0290). For identification of metabolites, relative migration times of all peaks were measured by normalization to the reference compound 3-aminopyrrolidine. Then, metabolites were identified by comparing their $m/z$ values and relative migration times to the metabolite standards. Metabolites were quantified by comparing peak areas to calibration curves generated using internal standardization techniques with methionine sulfone. For anionic compounds, a commercially available COSMO(+) (chemically coated with cationic polymer) capillary (50 mm i.d. × 105 cm) was used with a 50 mM ammonium acetate solution (pH 8.5) as the electrolyte (Nacalai Tesque, Kyoto, Japan). Methanol/5 mM ammonium acetate (50% v/v) containing 0.1 mM hexakis (2,2-difluoroethoxy) phosphazene was run as the sheath liquid at 10 mL/min. ESI-TOFMS was performed in negative ion mode, and the capillary voltage was set to 3.5 kV. For the anion analysis, trimesate and CAS were used for the reference of the internal standards, respectively. CE-TOFMS raw data were analyzed using Master Hands software (ver. 2.17.0.10). For each experiment, data conversion, binning data into 0.02 $m/z$ slices, baseline elimination, peak picking, integration, and elimination of redundant features were conducted to yield the possible peaks. Data matrices were aligned based on corrected migration times, and metabolite were assigned to the aligned peaks by matching m/z and the corrected migration times using standards metabolite library. Relative peak areas were calculated by the ratio of peak area divided by the internal standards. Metabolite concentrations were calculated based on the relative peak area between the sample and the standard.

**Pulse-chase analysis.** Dynamic pre-rRNA processing analysis was performed by pulse-chase labeling was performed[51]. Thirty minutes before the pulse labeling, HCT116 cells were washed in PBS and pre-incubated for 30 min in medium lacking methionine. The medium lacking methionine was either: nutrient deprivation (ND), or nutrient deprivation supplemented with glutamine (ND + Gln), or methionine-free DMEM (Gibco) with 10% dialyzed FBS (Sigma). Cells were then labeled with 5 μCi/ml L-(methyl-³H)-methionine (PerkinElmer) for 30 min. After labeling, the unincorporated radioactive methionine was chased by washing cells with PBS containing 0.3 mg/mL cold methionine (Sigma). Cells were then collected immediately (0-hr time point) or incubated for 1, 2 3, 4 h in their respective medium supplemented with 0.3 mg/ml cold methionine (Formedium) prior to collection. Total RNA was extracted and separated on denaturing agarose gels (6% formaldehyde/1.2% agarose in 50 mM HEPES-1mM EDTA) and transferred to nylon membranes (GeneScreen) for imaging on Fuji tritium screen (Fujifilm). After $\sim$2–3 weeks exposure, the signal was captured with a Phosphor-imager (FLA-7000; Fujifilm) and quantitated.

**Statistics and reproducibility.** Plotted values are shown as means ± SD throughout this study unless otherwise indicated. Indicated p values were obtained using one-way analysis of variance (ANOVA) followed by Student–Newman–Keuls multiple comparisons for post hoc test, paired Student's $t$ tests, or Mann–Whitney $U$ test. P values denoted as $*p < 0.05$; $**p < 0.01$; $***p < 0.001$ (ns, not significant). Representative results of two biological replicates were shown for: 2h, 3e, f, 4a, b, 5a, d, f, 6a, b, g, Supplementary 3a, b, 5b, d.

**Reporting summary.** Further information on research design is available in the Nature Research Reporting Summary linked to this article.

## Data availability

Source data are provided with this paper.

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

## Acknowledgements

We thank the members of the Laboratory for Systems Biology and Medicine and Integrative Nutriomics and Oncology, RCAST, the University of Tokyo. We especially thank Dr. Shintaro Iwasaki and Dr. T. Tanaka for helpful discussions. This work was supported by Grant-in-Aid for Scientific Research B (19H03496, T.O.), Scientific Research on Innovative Areas (20H04834, T.O.) and Grant-in-Aid for challenging Exploratory Research (19K22553, 21K19399, T.O.) from the Ministry of Education, Culture, Sports, Science and Technology of Japan, The Tokyo Biochemical Research Foundation (T.O.), Life Science Foundation of Japan (T.O.), Princess Takamatsu Cancer research Found (T.O.), The Naito Foundation (T.O.), The Uehara Memorial Foundation (T.O.).

## Author contributions

M.P. conceived the study; M.P., C.Z., K.I., M.S., M.K., M.N., A.O., T.I., T.So., T.O. performed experiments; T.O., H.Z., M.C., H.A., J.S., Y.M., T. Su., advised on critical experiments; M.P., D.L., C.G.P., T.O. wrote the manuscript.

## Competing interests

The authors declare no competing interests.
