## [Peer Review File · Nature Communications]

Glutamine deficiency in solid tumor cells confers resistance to ribosomal RNA synthesis inhibitorsEditorial Note: This manuscript has been previously reviewed at another journal that is not operating a transparent peer review scheme. This document only contains reviewer comments and rebuttal letters for versions considered at *Nature Communications*.

Reviewers' comments:

Reviewer #2 (Remarks to the Author):

It always find a bit puzzling to re-review a revised version of a manuscript that I previously rejected in another journal. But in this case, I think it is only fair to say that the authors did a good job replying to my comments. I think this manuscript can be published at Nat Commun.

Reviewer #3 (Remarks to the Author):

This manuscript reports that glutamine as the key nutrient that promotes the de novo protein synthesis of uL5 and uL18. Mechanistically, glutamine activates Ras/Raf/MEK/ERK signalling, which promotes uL5/uL18 synthesis through stimulating mTORC1 and eEF2. Pharmacologically inhibiting Raf/MEK/ERK inhibits the synthesis of uL5 and uL18 and abrogates the stabilization of p53. Depriving cancer cells of glutamine blocks the p53 nucleolar surveillance pathway, thus hampering the therapeutic response to Pol I inhibitors. While some of the findings in this manuscript are interesting, the results support the main conclusions in this manuscript are not strong and convincing. This manuscript is too premature to be published in this journal.

This manuscript has following major flaws.

1. Some important conclusions heavily depend on literature but not the results directly derived from this study. Given that different cells and different experimental conditions can easily lead to different observations, some important conclusions are not strong and convincing.
2. Some important results lack good and strict controls, and therefore, some results are not solid and strong to support the major conclusions of this manuscript. Some conclusions could be just association. Some conclusions are mainly supported by pharmacological approaches (which usually have off-target effects) but not genetic approaches. Particularly, the role of p53 in this stress response is not clearly demonstrated. The reviewer also does not agree with authors' comment that "We were unable to use siRNAs to knockdown Raf/MEK/ERK because siRNAs typically require 48-72 h for the knockdown to occur, and cancer cells do not survive for longer than ~36 h under ND so it was technically not feasible" since CRISPR/Cas9 approach which has been widely used can easily address this problem of siRNA. In addition, the role of MDM2 in this stress response has not been carefully investigated.
3. Authors did not sufficiently respond to many important critics raised by previous reviewers. Many important critics require further experiments and convincing experimental results but not only superficial discussion and simple citation of some other references.
4. The manuscript is not well written and presented. Some statements are not logical or accurate. Data are not well presented. For instance, only 4 figures are presented in the main text, whereas 7 supplementary figures are presented. These 4 figures do not provide sufficient information to support some major conclusions of the manuscript.

Reviewer #4 (Remarks to the Author):

In this study, M.S. Pan et al. showed that glutamine deprivation impairs the maturation of the 47S-pre rRNA but does not induce p53 activation by the uL5/uL18/5S rRNA potentially due to the inhibition of uL5 and uL18 translation. They show that pre-treatment with RNA Pol I inhibitors followed by glutamine refeeding induces p53-dependent apoptosis, mediated by uL5 and uL18. They propose that this is due to the failure to accumulate unmaturing pre-rRNAs that can no longer be monopolized during refeeding to resume ribosome biogenesis and incorporate uL5 and uL18 into the ribosome. This work is of potential interest as it proposes to shed light on the mechanisms of resistance of solid tumors to RNA pol I inhibitors, and could have direct implication in the clinic. However, many conclusions of the authors are based on indirect evidences that do not support the claims of the authors, and the findings are not convincingly demonstrated in solid tumors in-vivo. Therefore I do not think that this paper is suitable for publication in Nature Communications in its current form. Here are my major concerns:

1- Most of the findings rely on observations in cell lines in-vitro. Even though some experiments have been performed in-vivo, they provide only indirect, and ambiguous proofs that the findings of the authors are recapitulated in-vivo. First, the analysis of the stability of pre-rRNAs in the core/periphery of the tumors is based on the intratumoral injection of high dose Actinomycin D, and is compared to the stability of pre-rRNA of in-vitro treated cells. This comparison is not valid as it does not take in account the distribution and metabolism of the drug in-vivo, and at any time the authors verify that the drug hits the target in the tumor. Moreover, It's disturbing that the authors calculate the pre-rRNA half-life in tumors but only show the time corresponding to a 40% reduction (fig. S2B). Finally, they analyzed the global pools of pre-rRNAs by qPCR (without providing other basic controls such as housekeeping genes, 18S or 28S rRNA) when pre-rRNA species should have been analyzed separately by Northern-blot. Finally, the most critical compound that one is expecting in the metabolomic analysis of the core Vs periphery is glutamine, but this metabolite is absent from the list.

2- An important claim of the authors is that "failure to accumulate pre-rRNAs during stress leads to uL5/uL18-dependent activation and cell death upon nutrient restoration", but again, this statement is based on indirect findings that can be interpreted differently. This conclusion is based on the fact that a pre-treatment with RNA-pol I inhibitors, prevents de-novo transcription of 47S rRNA and therefore accumulation of downstream pre-rRNA transcripts. However, it is well established that inhibition of RNA pol I transcription lead to changes in nucleolar morphology, to the formation of nucleolar caps, and to the diffusion of nucleolar proteins in the nucleoplasm. How can the authors discard that one of this event is not responsible of the uL5/uL18-dependent activation under nutrient refeeding?

3- As "nucleolar stress" leads to the redirection of the complex made of nascent ribosomal proteins uL5, uL18 and 5S rRNA from the incorporation into the ribosome to the binding of Mdm2, the authors hypothesize that glutamine deprivation does not activate p53 because of the reduced translation of uL5 and uL18. However this remain hypothetical. For example, the authors do not show uL5/uL18 de-novo translation, co-immunoprecipitations of uL5/uL18 with Mdm2, or that p53 translation is not affected.

4- From the method and the legend, it is difficult to understand the authors have isolated polysome-bound RP-mRNAs from Figure 3e. This result is hard to interpret if the authors do not show the polysome profiles with the fractions they have isolated. Polysome profiles will also be important to understand how nutrient deprivation affects the pools of ribosomes and whether glutamine refeeding is sufficient to resume ribosome subunits synthesis and/or translation.

Reviewers' comments:

Reviewer #2 (Remarks to the Author):

It always find a bit puzzling to re-review a revised version of a manuscript that I previously rejected in another journal. But in this case, I think it is only fair to say that the authors did a good job replying to my comments. I think this manuscript can be published at Nat Commun.

Reviewer #3 (Remarks to the Author):

This manuscript reports that glutamine as the key nutrient that promotes the de novo protein synthesis of uL5 and uL18. Mechanistically, glutamine activates Ras/Raf/MEK/ERK signalling, which promotes uL5/uL18 synthesis through stimulating mTORC1 and eEF2. Pharmacologically inhibiting Raf/MEK/ERK inhibits the synthesis of uL5 and uL18 and abrogates the stabilization of p53. Depriving cancer cells of glutamine blocks the p53 nucleolar surveillance pathway, thus hampering the therapeutic response to Pol I inhibitors. While some of the findings in this manuscript are interesting, the results support the main conclusions in this manuscript are not strong and convincing. This manuscript is too premature to be published in this journal.

This manuscript has following major flaws.

1. Some important conclusions heavily depend on literature but not the results directly derived from this study.

With all due respect, we believe this statement is truly incorrect and simply unfair. As shown in the Table below, every important conclusion of our work is directly supported by experiments and clearly does NOT depend on literature. Table 1.1 systematically summarizes the manuscript's most central conclusions and the experiments that directly support our findings. On this basis, it is really not clear at all to any of the co-authors of this work, to which conclusion of ours this reviewer is referring to? Without any clearer description, this comment cannot be addressed any further.

Table 1.1

Conclusion	Cell lines used	Figure #
Nutrient deprivation upregulates pre-rRNA expression	HCT116, A549, U2OS, MKN45, 3T3-L1	1c, 1d, 1f
Nutrient deprivation slows pre-rRNA processing	HCT116, A375, A549, U2OS, MKN45	2a, 2c, 2d
Inhibiting pre-rRNA expression under nutrient deprivation leads to nutrient-induced apoptosis upon nutrient restoration	HCT116 A375	3f, 3g, 3h

Inhibiting pre-rRNA expression under nutrient deprivation activates the uL5/uL18-dependent nucleolar surveillance pathway upon metabolic recovery	HCT116 A375 A549	4b, 4c, 4d
Glutamine restoration activates p53 in pre-rRNA depleted cells. Glutamine restoration activates the ERK signalling pathway in pre-rRNA depleted cells. Inhibitors targeting Raf/MEK/ERK block the activation of p53 by glutamine.	HCT116 A375	5a, 5c, 5d, 5f, 5g
Glutamine deprivation inhibits the activation of p53 by CX-5461 and other Pol I inhibitors	HCT116, A375, A549, LNCaP, U2OS, MKN45	6a, 6b, 6c

Given that different cells and different experimental conditions can easily lead to different observations, some important conclusions are not strong and convincing.

We completely disagree. Our use of multiple experimental models/systems (mouse, human etc) in which we have perfectly reproduced our observations argues in favor of our conclusions making them stronger. We do not understand what point the referee is trying to make here.

2. Some important results lack good and strict controls, and therefore, some results are not solid and strong to support the major conclusions of this manuscript.

Again, this is a very generic comment aimed to harm the work. Without a suitable description of what such “so-called missing controls” could be, it is simply not possible for us to answer this.

By comparison: some additional controls were specifically requested at the first round of revision by referee #2. The requests of referee #2 were

very clear, and perfectly addressed by us to the full satisfaction of reviewer #2 (see Table 1.2).

If reviewer #3 wants more controls, it would be useful if these could be spelled out clearly so we can provide them immediately; that is, if they are indeed required.

Table 1.2

Requested by:	Figure	Additional controls added
Reviewer 1	3e	We blotted for additional ribosome-free r-proteins as an additional control
Reviewer 2	1c, 1d	We repeated the northern blotting to increase the intensity of the pre-rRNA bands in the control lanes
Reviewer 2	3i	We show that pre-rRNA synthesis resumes after drug washout as a control
Reviewer 2	4a	We show vehicle (PBS) treated cells as an additional control
Reviewer 2	5a	We show vehicle (PBS) treated cells as an additional control
Reviewer 2	Supplementary 5d	We show glutamine deprived (-Gln) cells as an additional control

Some conclusions could be just association.

We respectfully disagree. Our main conclusions are directly supported by experimental data (Table 1.1).

Some conclusions are mainly supported by pharmacological approaches (which usually have off-target effects) but not genetic approaches.

The approach we have developed in this work is perfectly in line with similar Raf/MEK/ERK papers published at the highest level. For instance, please refer to the work by Kinsey et al., who published a Ras/Raf/MEK/ERK paper to Nature Medicine without using genetic techniques (Nature Medicine 2019, PMID 30833748, Fig 1c-f).

We would like to remind reviewer #3 that we show that no less than six independent and widely used Raf/MEK/ERK inhibitors all produce the same inhibitory effect on the activation of p53 by glutamine. Remarkably, the six inhibitors we used target individual nodes of the same pathway (2

Raf inhibitors, 2 MEK inhibitors, and 2 ERK inhibitors). We believe this is bona fide evidence that supports our model.

[REDACTED]

[REDACTED]

Particularly, the role of p53 in this stress response is not clearly demonstrated.

Fully disagree: This manuscript clearly shows that p53 drives NR-mediated apoptosis in cells that fail to accumulate pre-rRNAs during starvation. For instance, Fig 4a shows that NR activates p53. Fig 4b, 4c shows that p53 is activated by uL5/uL18. Fig 4d shows that genetic ablation of p53 blocks NR-mediated apoptosis.

The reviewer also does not agree with authors' comment that "We were unable to use siRNAs to knockdown Raf/MEK/ERK because siRNAs typically require 48-72 h for the knockdown to occur, and cancer cells do not survive for longer than ~36 h under ND so it was technically not feasible" since CRISPR/Cas9 approach which has been widely used can easily address this problem of siRNA.

[REDACTED]

We have shown that six independent inhibitors targeting the ERK pathway all inhibit the metabolic activation of p53 by glutamine. These inhibitors

are widely used in Raf/MEK/ERK studies, and we have employed these drugs at published concentrations. Given our robust pharmacological data, and our new ERK siRNA depletion data, we believe that this point is addressed satisfactorily. Indeed, generating CRISPR-Cas9 knockout cell lines at this stage, which may take up to several months, is not feasible.

In addition, the role of MDM2 in this stress response has not been carefully investigated.

Our manuscript shows that the nucleolar surveillance pathway is activated in pre-rRNA-depleted cells during NR. The nucleolar surveillance pathway is defined by uL5/uL18 binding and inhibiting MDM2, which leads to p53 activation. We therefore immunoprecipitated MDM2 to examine the expression of associated uL5/uL18.

HCT116 cells were treated to ND+Vehicle or ND+CX-5461 for 24 hours, followed by NR (24 hours) (Fig 4b, new data). The cell lysates were subject to immunoprecipitation using control IgG or MDM2 primary antibodies. Input fraction: MDM2 was undetectable in starved control cells (ND), but was upregulated in pre-rRNA-depleted cells subject to NR (ND+CX-5461 to NR) (Fig 4b). IgG immunoprecipitation fraction: uL5 and uL18 was not detected in the control IP (Fig 4b). MDM2 immunoprecipitation: NR increased the expression of MDM2-bound uL5/uL18 in pre-rRNA-depleted cells (Fig 4b).

3. Authors did not sufficiently respond to many important critics raised by previous reviewers. Many important critics require further experiments and convincing experimental results but not only superficial discussion and simple citation of some other references.

Again a generic comment that cannot be answered specifically, apart from mentioning that Reviewer #2 who initially raised criticisms, was very well convinced by our revision.

4. The manuscript is not well written and presented. Some statements are not logical or accurate. Data are not well presented. For instance, only 4 figures are presented in the main text, whereas 7 supplementary figures are presented. These 4 figures do not provide sufficient information to support some major conclusions of the manuscript.

Thank you. We deeply apologize for this. The revised manuscript now contains 6 main figures. We have also sent the manuscript to a professional scientific editing service.

Reviewer #4 (Remarks to the Author):

In this study, M.S. Pan et al. showed that glutamine deprivation impairs the maturation of the 47S-pre rRNA but does not induce p53 activation by the uL5/uL18/5S rRNA potentially due to the inhibition of uL5 and uL18 translation. They show that pre-treatment with RNA Pol I inhibitors followed by glutamine refeeding induces p53-dependent apoptosis, mediated by uL5 and uL18. They propose that this is due to the failure to accumulate unmaturing pre-rRNAs that can no longer be monopolized during refeeding to resume ribosome biogenesis and incorporate uL5 and uL18 into the ribosome. This work is of potential interest as it proposes to shed light on the mechanisms of resistance of solid tumors to RNA pol I inhibitors, and could have direct implication in the clinic. However, many conclusions of the authors are based on indirect evidences that do not support the claims of the authors, and the findings are not convincingly demonstrated in solid tumors in-vivo. Therefore I do not think that this paper is suitable for publication in Nature Communications in its current form. Here are my major concerns:

1- Most of the findings rely on observations in cell lines in-vitro. Even though some experiments have been performed in-vivo, they provide only indirect, and ambiguous proofs that the findings of the authors are recapitulated in-vivo. First, the analysis of the stability of pre-rRNAs in the core/periphery of the tumors is based on the intratumoral injection of high dose Actinomycin D, and is compared to the stability of pre-RNA of in-vitro treated cells. This comparison is not valid as it does not take in account the distribution and metabolism of the drug in-vivo, and at any time the authors verify that the drug hits the target in the tumor.

[REDACTED]

We note that it only takes ~10 nM actinomycin D to inhibit Pol I *in vitro*, but our solution of actinomycin D was 1.3 mM (1300000 nM). In order to verify the on-target activity of actinomycin D *in vivo*, we examined the nucleoli of control and treated tumors using nucleolin as a marker. HCT116 tumors were intratumorally injected with actinomycin D for 30 minutes, and tissues were preserved with OCT compound for cryosectioning and immunofluorescence. We found many nucleolin-positive nucleoli in control tumors (Supplementary Fig. 2a, new data). In striking contrast, tumors that were injected with actinomycin D had no identifiable nucleoli; the protein was redistributed throughout the cell and was mostly detected in the cytoplasm (Supplementary Fig. 2a).

We also measured the concentration of actinomycin D in tumoral peripheral and core samples using mass spectrometry.

[REDACTED]

[REDACTED]

Moreover, It's disturbing that the authors calculate the pre-rRNA half-life in tumors but only show the time corresponding to a 40% reduction (fig. S2B). Finally, they analyzed the global pools of pre-RNAs by qPCR (without providing other basic controls such as housekeeping genes, 18S or 28S rRNA) when pre-RNA species should have been analyzed separately by Northern-blot.

Thank you. We agree. We have repeated this experiment and have extended the time course of analysis from 2 to 8 hours (Fig 2i, new data). Pre-rRNAs from the periphery of the tumors displayed a half-life of approximately ~3.5 hours while those from the core are characterized by a half-life of ~8.5 hours. Tumor-bearing mice were severely lethargic after 8 h so the experiment was stopped at this time point.

We also tried to repeat this experiment using Northern-blotting instead of qPCR (Reviewers Only Fig 2.3). However, we were unable to detect any differences in pre-rRNA-depletion between tumor periphery and core samples. This is in contrast to the qPCR experiment, which may possibly be explained by differences in assay sensitivities. We also faced technical issues when performing this experiment. Unfortunately, the RNA samples sent from Tokyo to Belgium for NB analysis were partially degraded during transport.

[REDACTED]

[REDACTED]

Finally, the most critical compound that one is expecting in the metabolomic analysis of the core Vs periphery is glutamine, but this metabolite is absent from the list.

Agree. Thank you. The metabolomics experiment analyzed periphery/core tissues collected from 5 independent tumors (“Tumors #1-5”). Each periphery/core sample was further divided into two pieces and analyzed in duplicate.

[REDACTED]

Much to our satisfaction, several key nutrients were significantly decreased in core tissues vs. periphery, including the most important glutamine, as well as glucose 1-phosphate, arginine, and serine (Supplementary Fig. 2b, new data). We would like to thank the reviewer for this comment, which has allowed us to clarify this point.

[REDACTED]

2- An important claim of the authors is that “failure to accumulate pre-rRNAs during stress leads to uL5/uL18-dependent activation and cell death upon nutrient restoration”, but again, this statement is based on indirect findings that can be interpreted differently. This conclusion is based on the fact that a pre-treatment with RNA-pol I inhibitors, prevents de-novo transcription of 47S rRNA and therefore accumulation of downstream pre-rRNA transcripts. However, it is well established that inhibition of RNA pol I transcription lead to changes in nucleolar morphology, to the formation of nucleolar caps, and to the diffusion of nucleolar proteins in the nucleoplasm. How can the authors discard that one of this event is not responsible of the uL5/uL18-dependent activation under nutrient refeeding?

The nucleolar surveillance pathway occurs as a chain of events: inhibition of RNA Pol I → depletion of pre-rRNA intermediates → nucleolar disruption → uL5/uL18-mediated activation of p53. Pre-rRNAs form the primary scaffold upon which nucleolar proteins and pre-ribosomal particles bind to, meaning that nucleolar morphology is dependent on nucleolar RNA levels (PMID: 27474438). The importance of pre-rRNAs to nucleolar morphology is further highlighted by the finding that impaired pre-rRNA processing causes unprocessed rRNAs to accumulate, which in turn leads to higher nucleolar RNA content and enlarged nucleoli (PMID:

25732822). Thus, nucleolar morphology is dependent on nucleolar RNA abundance. We believe that the Reviewer is correct on the role of nucleolar disruption on uL5/uL18-mediated p53 activation. However, since pre-rRNA depletion precedes nucleolar disruption, we believe that the Reviewer is re-stating our conclusion in a different manner.

We present three lines of evidence that show that the activation of p53 by uL5/uL18 begins with pre-rRNA depletion rather than nucleolar reorganization.

Summarized:

1. Nucleolar disruption by Pol I inhibitors is transient and reversible
2. Transient nucleolar disruption does not cause newly synthesized uL5/uL18 to diffuse to the nucleoplasm
3. Pre-rRNA depletion by CX-5461 is required for p53 activation

[REDACTED]

[REDACTED]

[REDACTED]

After 24 h, cells were subject to NR for 1 h. Immunofluorescence was performed on nucleolin, UBF, and uL18.

2. Transient nucleolar disruption does not cause newly synthesized uL5/uL18 to diffuse to the nucleoplasm

Our manuscript proposes that nutrient refeeding restores global protein synthesis, causing newly translated uL5/uL18 to activate p53. We next tested whether newly translated uL5/uL18 can localize to the nucleolus after transient nucleolar disruption. HCT116 cells were pre-treated to CX-5461/BMH-21 for 1 hour, and uL5-FLAG or uL18-FLAG was subsequently expressed in the pre-treated cells (in the presence of the drug). We chose to specifically express uL5-FLAG/uL18-FLAG after CX-5461/BMH-21 pre-treatment in order to examine the localization of newly synthesized uL5/uL18 after nucleolar segregation.

[REDACTED]

[REDACTED]

3. Pre-rRNA depletion by CX-5461 is required for p53 activation

[REDACTED]

[REDACTED]

[REDACTED]

[REDACTED]

[REDACTED]

In conclusion, we present three lines of evidence that indicate that the activation of p53 upon nutrient refeeding is not due to nucleolar disintegration: (1) the effects of nucleolar disruption by CX-5461/BMH-21 are transient and reversible, (2) newly synthesized uL5/uL18 can successfully localize to the nucleolus after transient nucleolar disruption, and (3) the activation of p53 via uL5/uL18 requires pre-rRNA-depletion.

3- As “nucleolar stress” leads to the redirection of the complex made of nascent ribosomal proteins uL5, uL18 and 5S rRNA from the incorporation into the ribosome to the binding of Mdm2, the authors hypothesize that glutamine deprivation does not activate p53 because of the reduced translation of uL5 and uL18. However this remain hypothetical. For example, the authors do not show uL5/uL18 de-novo translation, co-immunoprecipitations of uL5/uL18 with Mdm2, or that p53 translation is not affected.

[REDACTED]

[REDACTED]

[REDACTED]

[REDACTED]

4- From the method and the legend, it is difficult to understand the authors have isolated polysome-bound RP-mRNAs from Figure 3e. This result is hard to interpret if the authors do not show the polysome profiles with the fractions they have isolated. Polysome profiles will also be important to understand how nutrient deprivation affects the pools of ribosomes and whether glutamine refeeding is sufficient to resume ribosome subunits synthesis and/or translation.

[REDACTED]

[REDACTED]

REVIEWERS' COMMENTS

Reviewer #3 (Remarks to the Author):

This revised manuscript does not sufficiently address the reviewer's comments. Overall this study was not well designed or performed. Some important concepts (such as CX-05461 mainly functions as a polymerase I inhibitor) are not correct or inaccurate. The results don't strongly support the main conclusions of this manuscript. Some of the major flaws of this manuscript are listed as follows. Based on these major flaws, this manuscript is too premature to be considered for publication in this journal. Therefore, the reviewer suggest the rejection of this manuscript.

Major flaws:

1. CX-5461, which was reported to be a polymerase I (Pol I) inhibitor, was used as a major model in this study to block rRNA biogenesis in cells, leading to p53 stabilization. However, authors obviously missed some very important recent publications in the field including following two in PNAS and Nature Communications that have clearly shown that CX-5461 primarily targets topoisomerase II (TOP2) to induce DNA damages in cells.

a. The primary mechanism of cytotoxicity of the chemotherapeutic agent CX-5461 is topoisomerase II poisoning. Bruno P. et al. Proc Natl Acad Sci U S A. 2020 Feb 25;117(8):4053-4060.

b. The chemotherapeutic CX-5461 primarily targets TOP2B and exhibits selective activity in high-risk neuroblastoma Pan M. et al. Nat Commun. 2021; 12(1):6468.

Given that CX-5461 mainly targets TOP2 (to directly induce DNA damage) instead of Pol I (to induce ribosomal biogenesis), the results and the major conclusions in this manuscript based on the previous (wrong) concept that CX-5461 mainly blocks rRNA biogenesis are questionable or could be very wrong. Also, it is well established in the field that many other Top2 inhibitors (such as etoposide and doxorubicin) activate p53 through directly inducing DNA damages. Therefore, CX-5461 can directly activate p53 through inducing DNA damages as a major mechanism. The conclusion that p53 activation by CX-5461 through blocking ribosome biogenesis and through uL5/uL18 and MDM2 interactions could be wrong or not a major mechanism. Authors should repeat the experiments performed in the above-mentioned two articles to prove that CX-5461 can inhibit TOP2 to cause DNA damage as a major mechanism to activate p53.

2. Authors cited references that p53 activation in their system is through uL5/uL18 and MDM2. However, authors did not provide any direct evidence. Given that p53 activation is highly cell, tissue, and stress dependent, authors have to show MDM2 knockout or deficiency will block p53 activation by CX-5461. Considering that CX-5461 can inhibit TOP2 to cause DNA damage to activate p53, the reviewer doubt that this p53 regulation is mainly through uL5/uL18 and MDM2 interactions.

3. Several studies have clearly demonstrated that glutamine starvation can activate p53 in cells. (Michael A Reid, et al. The B55 α subunit of PP2A drives a p53-dependent metabolic adaptation to glutamine deprivation. Mol Cell. 2013 Apr 25;50(2):200-11. Lowman XH et al. p53 Promotes Cancer Cell Adaptation to Glutamine Deprivation by Upregulating Slc7a3 to Increase Arginine Uptake. Cell Rep. 2019 Mar 12;26(11):3051-3060. IKK β activates p53 to promote cancer cell adaptation to glutamine deprivation. Mari B Ishak Gabra et al. Oncogenesis. 2018 Nov 26;7(11):93.)

The main conclusion on p53 regulation in these published studies are inconsistent with the results in this manuscript showing that glutamine restoration activates p53. Authors should repeat their experiments in different cell lines and using different conditions to understand and explain the inconsistency between their observations and previous publications.

4. Experiments in this manuscript were mainly performed in cell lines. Very limited studies were performed in vivo using xenograft tumors. Considering the culture conditions in vitro are very different from that in in vivo, the findings from cultured cells could be very different from the "solid tumors"

(authors used the title “Glutamine deficiency in solid tumors confers resistance to Ribosomal RNA synthesis inhibitors”). It is crucial to understand the role and mechanism in vivo and in solid tumors. Authors should repeat main findings in cultured cell lines using xenograft tumors. In addition, p53 knockout mouse is well developed and commercial available, which should be used to repeat important experiments performed in cultured cell lines to see if these results can be observed in vivo to enhance the physiological relevance of this study.

5. Authors only used p53 wild-type cell lines for xenograft tumors. No matched p53^{-/-} cell lines were used for xenograft tumors. Although p53 activation was observed in tumors, authors did not provide direct evidence that the observed results are p53 dependent.

Minor concerns:

1. Fig 2A: it appears that the results of the Control group were not from the same gels of two treated groups. It is difficult to evaluate results from different gels.

Reviewer #4 (Remarks to the Author):

The work done by M.Pan et al. has addressed most of my concerns, even though some technical problems persist (no Northern Blot supporting the qPCR of Fig2i, metabolomic data perform on 2 tumors only, no incorporation of uL5-FLAG or uL18-FLAG into the cytoplasmic ribosomes). However, the answers of the authors raise 2 major issues:

1- In response to my point 3 (Reviewer 4.3, figures to authors 2.19), the authors conclude that the expression of nascent uL18 and uL5 is not modified by Glutamine deprivation (Quantifications are missing, see point 2). However, in the manuscript the authors still state that “Glutamine metabolically activates p53 by inducing uL5/uL18 translation”. The authors should at least rewrite the manuscript according to their new results, and discuss an alternative mechanism

2- I don't understand why most of the new results have not been included in the manuscript or supplementaries (21 figures). In my opinion they are important results and should be included. This is especially the case for Figures 2.2, 2.7, 2.8 ; 2.11 to 2.14 ; 2.16 to 2.20. For 2.18 and 2.19, additional controls (Streptavidin IP without AHA, other ribosomal and non-ribosomal proteins) as well as quantifications should be included.

REVIEWER #3

This revised manuscript does not sufficiently address the reviewer's comments. Overall this study was not well designed or performed. Some important concepts (such as CX-05461 mainly functions as a polymerase I inhibitor) are not correct or inaccurate. The results don't strongly support the main conclusions of this manuscript. Some of the major flaws of this manuscript are listed as follows. Based on these major flaws, this manuscript is too premature to be considered for publication in this journal. Therefore, the reviewer suggest the rejection of this manuscript.

Major flaws:

1. CX-5461, which was reported to be a polymerase I (Pol I) inhibitor, was used as a major model in this study to block rRNA biogenesis in cells, leading to p53 stabilization. However, authors obviously missed some very important recent publications in the field including following two in PNAS and Nature Communications that have clearly shown that CX-5461 primarily targets topoisomerase II (TOP2) to induce DNA damages in cells.

a. The primary mechanism of cytotoxicity of the chemotherapeutic agent CX-5461 is topoisomerase II poisoning. Bruno P. et al. Proc Natl Acad Sci U S A. 2020 Feb 25;117(8):4053-4060.

b. The chemotherapeutic CX-5461 primarily targets TOP2B and exhibits selective activity in high-risk neuroblastoma Pan M. et al. Nat Commun. 2021; 12(1):6468.

Given that CX-5461 mainly targets TOP2 (to directly induce DNA damage) instead of Pol I (to induce ribosomal biogenesis), the results and the major conclusions in this manuscript based on the previous (wrong) concept that CX-5461 mainly blocks rRNA biogenesis are questionable or could be very wrong. Also, it is well established in the field that many other Top2 inhibitors (such as etoposide and doxorubicin) activate p53 through directly inducing DNA damages. Therefore, CX-5461 can directly activate p53 through inducing DNA damages as a major mechanism. The conclusion that p53 activation by CX-5461 through blocking ribosome biogenesis and through uL5/uL18 and MDM2 interactions could be wrong or not a major mechanism. Authors should repeat the experiments performed in the above-mentioned two articles to prove that CX-5461 can inhibit TOP2 to cause DNA damage as a major mechanism to activate p53.

When we started this project, CX-5461 was, at the time, the most specific inhibitor of RNA Pol I with a defined mechanism of action involving targeting SL1. Moreover, to address possible off-target effects of CX-5461, we included the use of BMH-21. Nonetheless we have added these points to the discussion.

2. Authors cited references that p53 activation in their system is through uL5/uL18 and MDM2. However, authors did not provide any direct evidence. Given that p53 activation is highly cell, tissue, and stress dependent, authors have to show MDM2 knockout or deficiency will block p53 activation by CX-5461. Considering that CX-5461 can inhibit TOP2 to cause DNA damage to activate p53, the reviewer doubt that this p53 regulation is mainly through uL5/uL18 and MDM2 interactions.

We have already provided clear and direct evidence that p53 activation by NR is dependent on uL5/uL18. We are puzzled as to how “MDM2 knockout or deficiency” experiments will “block p53 activation”, as MDM2 is a negative regulator of p53.

3. Several studies have clearly demonstrated that glutamine starvation can activate p53 in cells. (Michael A Reid, et al. The B55 α subunit of PP2A drives a p53-dependent metabolic adaptation to glutamine deprivation. Mol Cell. 2013 Apr 25;50(2):200-11.

Lowman XH et al. p53 Promotes Cancer Cell Adaptation to Glutamine Deprivation by Upregulating Slc7a3 to Increase Arginine Uptake. Cell Rep. 2019 Mar 12;26(11):3051-3060.
IKK β activates p53 to promote cancer cell adaptation to glutamine deprivation. Mari B Ishak Gabra et al. Oncogenesis. 2018 Nov 26;7(11):93.)

The main conclusion on p53 regulation in these published studies are inconsistent with the results in this manuscript showing that glutamine restoration activates p53. Authors should repeat their experiments in different cell lines and using different conditions to understand and explain the inconsistency between their observations and previous publications.

These cited studies do not show that glutamine starvation can increase total p53 levels. Therefore, there are no inconsistencies between published literature and our study.

4. Experiments in this manuscript were mainly performed in cell lines. Very limited studies were performed in vivo using xenograft tumors. Considering the culture conditions in vitro are very different from that in vivo, the findings from cultured cells could be very different from the “solid tumors” (authors used the title “Glutamine deficiency in solid tumors confers resistance to Ribosomal RNA synthesis inhibitors”). It is crucial to understand the role and mechanism in vivo and in solid tumors. Authors should repeat main findings in cultured cell lines using xenograft tumors. In addition, p53 knockout mouse is well developed and commercially available, which should be used to repeat important experiments performed in cultured cell lines to see if these results can be observed in vivo to enhance the physiological relevance of this study.

Unfortunately, at this stage, we cannot provide further in vivo validation in addition to what we have already performed. This is not compatible with the time frame of the publication of this work. Our in vivo data already includes the use of two cancer cell lines of unrelated tumor origins, HCT116 and A375, with the use of AMG 232 as a positive control for p53 stabilization in vivo. We believe that this robust data entirely justifies our in vivo conclusions.

5. Authors only used p53 wild-type cell lines for xenograft tumors. No matched p53 $^{-/-}$ cell lines were used for xenograft tumors.

Our results show that CX-5461 fails to stabilize p53 in solid xenograft tumors, hence the need for p53 wild-type cell lines. Using p53 $^{-/-}$ tumors would not address this hypothesis.

Although p53 activation was observed in tumors, authors did not provide direct evidence that the observed results are p53 dependent.

Indeed, p53 activation was observed in tumors treated with the MDM2 inhibitor AMG 232. The anti-tumor activity of AMG 232 is strictly through p53 activation, thus providing evidence that the observed results are p53 dependent.

Minor concerns:

1. Fig 2A: it appears that the results of the Control group were not from the same gels of two treated groups. It is difficult to evaluate results from different gels.

The uncropped gel is now available in the Source Data.

REVIEWER #4

The work done by M.Pan et al. has addressed most of my concerns, even though some technical problems persist (no Northern Blot supporting the qPCR of Fig2i, metabolomic data perform on 2 tumors only, no incorporation of uL5-FLAG or uL18-FLAG into the cytoplasmic ribosomes).

However, the answers of the authors raise 2 major issues:

1- In response to my point 3 (Reviewer 4.3, figures to authors 2.19), the authors conclude that the expression of nascent uL18 and uL5 is not modified by Glutamine deprivation (Quantifications are missing, see point 2).

However, in the manuscript the authors still state that “Glutamine metabolically activates p53 by inducing uL5/uL18 translation”. The authors should at least rewrite the manuscript according to their new results, and discuss an alternative mechanism

We have now added this to the manuscript’s discussion: “Notably, glutamine deprivation did not decrease uL5/uL18 protein synthesis, indicating that the mechanism of p53 inhibition is independent of uL5/uL18 mRNA translation”

2- I don’t understand why most of the new results have not been included in the manuscript or supplementaries (21 figures). In my opinion they are important results and should be included. This is especially the case for Figures 2.2, 2.7, 2.8 ; 2.11 to 2.14 ; 2.16 to 2.20. For 2.18 and 2.19, additional controls (Streptavidin IP without AHA, other ribosomal and non-ribosomal proteins) as well as quantifications should be included.

Following the reviewer’s suggestion, we have now included Fig 2.2, 2.7, 2.8, 2.11-2.14, and 2.16-2.20 in the manuscript.

Fig 2.2 is shown as Supplementary 1e

Fig 2.7 & 2.8 is shown as Supplementary 3a (lower panel)

Fig 2.11-2.14 and Fig 2.16-2.20 is shown as Supplementary Fig 3c, 6a, 6b, 6d, 6e, 6f, 6g, and 6h.